# Understanding the Latent Space of Diffusion Models through the Lens of Riemannian Geometry

**Yong-Hyun Park**[*1], **Mingi Kwon**[*2], **Jaewoong Choi**[3], **Junghyo Jo**[†1], **Youngjung Uh**[†2],

[1]Seoul National University  [2]Yonsei University  [3]Korea Institute for Advanced Study

## Abstract

Despite the success of diffusion models (DMs), we still lack a thorough understanding of their latent space. To understand the latent space $\mathbf{x}_t \in \mathcal{X}$, we analyze them from a geometrical perspective. Our approach involves deriving the local latent basis within $\mathcal{X}$ by leveraging the pullback metric associated with their encoding feature maps. Remarkably, our discovered local latent basis enables image editing capabilities by moving $\mathbf{x}_t$, the latent space of DMs, along the basis vector at specific timesteps. We further analyze how the geometric structure of DMs evolves over diffusion timesteps and differs across different text conditions. This confirms the known phenomenon of coarse-to-fine generation, as well as reveals novel insights such as the discrepancy between $\mathbf{x}_t$ across timesteps, the effect of dataset complexity, and the time-varying influence of text prompts. To the best of our knowledge, this paper is the first to present image editing through $\mathbf{x}$-space traversal, editing only once at specific timestep $t$ without any additional training, and providing thorough analyses of the latent structure of DMs. The code to reproduce our experiments can be found at https://github.com/enkeejunior1/Diffusion-Pullback.

## 1 Introduction

The diffusion models (DMs) are powerful generative models that have demonstrated impressive performance [22, 51, 52, 17, 37]. DMs have shown remarkable applications, including text-to-image synthesis [45, 46, 5, 36], inverse problems [14, 31], and image editing [21, 54, 39, 35].

Despite their achievements, the research community lacks a comprehensive understanding of the latent space of DMs and its influence on the generated results. So far, the completely diffused images are considered as latent variables but it does not have useful properties for controlling the results. For example, traversing along a direction from a latent produces weird changes in the results. Fortunately, Kwon et al. [26] consider the intermediate feature space of the diffusion kernel, referred to as $\mathcal{H}$, as a semantic latent space and show its usefulness on controlling generated images. In the similar sense, some works investigate the feature maps of the self-attention or cross-attention operations for controlling the results [21, 54, 39], improving sample quality [8], or downstream tasks such as semantic segmentation [32, 55].

Still, the structure of the space $\mathcal{X}_t$ where latent variables $\{\mathbf{x}_t\}$ live remains unexplored despite its crucial role in understanding DMs. It is especially challenging because 1) the model is trained to estimate the forward noise which does not depend on the input, as opposed to other typical supervisions such as classification or similarity, and 2) there are lots of latent variables over multiple recursive timesteps. In this paper, we aim to analyze $\mathcal{X}$ in conjunction with its corresponding

---

[*]Equal Contribution    [†] Corresponding authors

37th Conference on Neural Information Processing Systems (NeurIPS 2023).

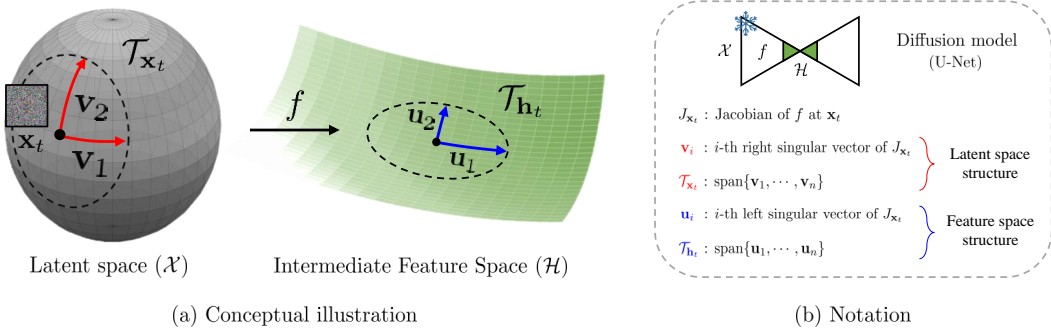

| | |
|---|---|
| (a) Conceptual illustration | (b) Notation |

Figure 1: **Conceptual illustration of local geometric structure.** (a) The local basis $\{\mathbf{v}_1, \mathbf{v}_2, \cdots\}$ of the local latent subspace $\mathcal{T}_{\mathbf{x}_t}$ within the latent space $\mathcal{X}$ is interlinked with the local basis $\{\mathbf{u}_1, \mathbf{u}_2, \cdots\}$ of the local tangent space $\mathcal{T}_{\mathbf{h}_t}$ in the feature space $\mathcal{H}$. (b) The derivation of these local bases is facilitated through the singular value decomposition (SVD) of the Jacobian, which emanates from the U-Net responsible for encoding the feature map $f$, linking $\mathcal{X}$ and $\mathcal{H}$.

representation $\mathcal{H}$, by incorporating a local geometry to $\mathcal{X}$ using the concept of a *pullback metric* in Riemannian geometry.

First, we discover the local latent basis for $\mathcal{X}$ and the corresponding local tangent basis for $\mathcal{H}$. The local basis is obtained by performing singular value decomposition (SVD) of the Jacobian of the mapping from $\mathcal{X}$ to $\mathcal{H}$. To validate the discovered local latent basis, we demonstrate that walking along the basis can edit real images in a semantically meaningful way. Furthermore, we can use the discovered local latent basis vector to edit other samples by using parallel transport, when they exhibit comparable local geometric structures. Note that existing editing methods manipulate the self-attention map or cross-attention map over multiple timesteps [21, 54, 39]. On the other hand, we manipulate only $\mathbf{x}_t$ once at a specific timestep.

Second, we investigate how the latent structures differ across different timesteps and samples as follows. The frequency domain of the local latent basis shifts from low-frequency to high-frequency along the generative process. We explicitly confirm it using power spectral density analysis. The difference between local tangent spaces of different samples becomes larger along the generative process. The local tangent spaces at various diffusion timesteps are similar to each other if the model is trained on aligned datasets such as CelebA-HQ or Flowers. However, this homogeneity does not occur on complex datasets such as ImageNet.

Finally, we examine how the prompts affect the latent structure of text-to-image DMs as follows. Similar prompts yield similar latent structures. Specifically, we find a positive correlation between the similarity of prompts and the similarity of local tangent spaces. The influence of text on the local tangent space becomes weaker along the generative process.

Our work examines the geometry of $\mathcal{X}$ and $\mathcal{H}$ using Riemannian geometry. We discover the latent structure of $\mathcal{X}$ and how it evolves during the generative process and is influenced by prompts. This geometric exploration deepens our understanding of DMs.

## 2 Related works

**Diffusion Models.**   Recent advances in DMs make great progress in the field of image synthesis and show state-of-the-art performance [50, 22, 51]. An important subject in the diffusion model is the introduction of gradient guidance, including classifier-free guidance, to control the generative process [17, 47, 4, 30, 36, 46]. The work by Song et al. [52] has facilitated the unification of DMs with score-based models using SDEs, enhancing our understanding of DMs as a reverse diffusion process. However, the latent space is still largely unexplored, and our understanding is limited.

**The study of latent space in GANs.**   The study of latent spaces has gained significant attention in recent years. In the field of Generative Adversarial Networks (GANs), researchers have proposed various methods to manipulate the latent space to achieve the desired effect in the generated images

[44, 41, 1, 20, 49, 59, 38]. More recently, several studies [60, 10] have examined the geometrical properties of latent space in GANs and utilized these findings for image manipulations. These studies bring the advantage of better understanding the characteristics of the latent space and facilitating the analysis and utilization of GANs. In contrast, the latent space of DMs remains poorly understood, making it difficult to fully utilize their capabilities.

**Image manipulation in DMs.**  Early works include Choi et al. [11] and Meng et al. [33] have attempted to manipulate the resulting images of DMs by replacing latent variables, allowing the generation of desired random images. However, due to the lack of semantics in the latent variables of DMs, current approaches have critical problems with semantic image editing. Alternative approaches have explored the potential of using the feature space within the U-Net for semantic image manipulation. For example, Kwon et al. [26] have shown that the bottleneck of the U-Net, $\mathcal{H}$, can be used as a semantic latent space. Specifically, they used CLIP [43] to identify directions within $\mathcal{H}$ that facilitate genuine image editing. Baranchuk et al. [6] and Tumanyan et al. [54] use the feature map of the U-Net for semantic segmentation and maintaining the structure of generated images. Unlike previous works, our editing method finds the editing direction without supervision, and directly traverses the latent variable along the latent basis.

**Riemannain Geometry.**  Some studies have applied Riemannian geometry to analyze the latent spaces of deep generative models, such as Variational Autoencoders (VAEs) and GANs [2, 48, 9, 3, 27, 28, 57]. Shao et al. [48] proposed a pullback metric on the latent space from image space Euclidean metric to analyze the latent space's geometry. This method has been widely used in VAEs and GANs because it only requires a differentiable map from latent space to image space. However, no studies have investigated the geometry of latent space of DMs utilizing the pullback metric.

## 3 Discovering the latent basis of DMs

In this section, we explain how to extract a latent structure of $\mathcal{X}$ using differential geometry. First, we introduce a key concept in our method: the *pullback metric*. Next, by adopting the local Euclidean metric of $\mathcal{H}$ and utilizing the pullback metric, we discover the local latent basis of the $\mathcal{X}$. Moreover, although the direction we found is *local*, we show how it can be applied to other samples via parallel transport. Finally, we introduce **x**-space guidance for editing data in the $\mathcal{X}$ to enhance the quality of image generation.

### 3.1 Pullback metric

We consider a curved manifold, $\mathcal{X}$, where our latent variables $\mathbf{x}_t$ exist. The differential geometry represents $\mathcal{X}$ through patches of tangent spaces, $\mathcal{T}_\mathbf{x}$, which are vector spaces defined at each point $\mathbf{x}$. Then, all the geometrical properties of $\mathcal{X}$ can be obtained from the inner product of $||d\mathbf{x}||^2 = \langle d\mathbf{x}, d\mathbf{x} \rangle_\mathbf{x}$ in $\mathcal{T}_\mathbf{x}$. However, we do not have any knowledge of $\langle d\mathbf{x}, d\mathbf{x} \rangle_\mathbf{x}$. It is definitely not a Euclidean metric. Furthermore, samples of $\mathbf{x}_t$ at intermediate timesteps of DMs include inevitable noise, which prevents finding semantic directions in $\mathcal{T}_\mathbf{x}$.

Fortunately, Kwon et al. [26] observed that $\mathcal{H}$, defined by the bottleneck layer of the U-Net, exhibits local linear structure. This allows us to adopt the Euclidean metric on $\mathcal{H}$. In differential geometry, when a metric is not available on a space, *pullback metric* is used. If a smooth map exists between the original metric-unavailable domain and a metric-available codomain, the pullback metric is used to measure the distances in the domain space. Our idea is to use the pullback Euclidean metric on $\mathcal{H}$ to define the distances between the samples in $\mathcal{X}$.

DMs are trained to infer the noise $\epsilon_t$ from a latent variable $\mathbf{x}_t$ at each diffusion timestep $t$. Each $\mathbf{x}_t$ has a different internal representation $\mathbf{h}_t$, the bottleneck representation of the U-Net, at different $t$'s. The differentiable map between $\mathcal{X}$ and $\mathcal{H}$ is denoted as $f : \mathcal{X} \rightarrow \mathcal{H}$. Hereafter, we refer to $\mathbf{x}_t$ as $\mathbf{x}$ for brevity unless it causes confusion. It is important to note that our method can be applied at any timestep in the denoising process. We consider a linear map, $\mathcal{T}_\mathbf{x} \rightarrow \mathcal{T}_\mathbf{h}$, between the domain and codomain tangent spaces. This linear map can be described by the *Jacobian* $J_\mathbf{x} = \nabla_\mathbf{x} \mathbf{h}$ which determines how a vector $\mathbf{v} \in \mathcal{T}_\mathbf{x}$ is mapped into a vector $\mathbf{u} \in \mathcal{T}_\mathbf{h}$ by $\mathbf{u} = J_\mathbf{x} \mathbf{v}$.

Using the local linearity of $\mathcal{H}$, we assume the metric, $||d\mathbf{h}||^2 = \langle d\mathbf{h}, d\mathbf{h} \rangle_\mathbf{h} = d\mathbf{h}^\top d\mathbf{h}$ as a usual dot product defined in the Euclidean space. To assign a geometric structure to $\mathcal{X}$, we use the pullback

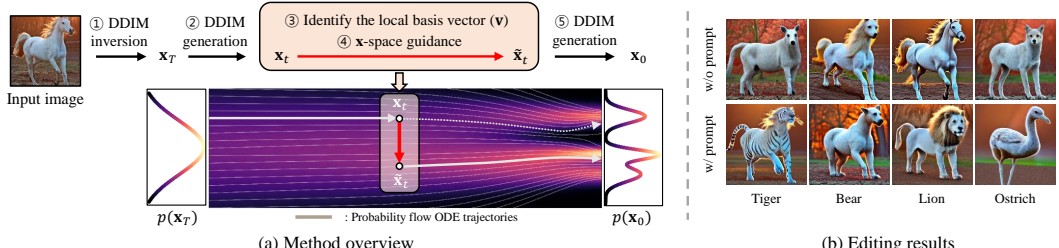

Figure 2: **Image editing with the discovered latent basis.** (a) Schematic depiction of our image editing procedure. ① An input image is subjected to DDIM inversion, resulting in an initial noisy sample $\mathbf{x}_T$. ② The sample $\mathbf{x}_T$ is progressively denoised until reaching the point $t$ through DDIM generation. ③ Subsequently, the local latent basis $\{\mathbf{v}_1, \cdots, \mathbf{v}_n\}$ is identified by using the pullback metric. ④ This enables the manipulation of the sample $\mathbf{x}_t$ along one of the basis vectors using $\mathbf{x}$-space guidance. ⑤ The DDIM generation concludes with the progression from the modified latent variable $\tilde{\mathbf{x}}_t$. (b) Examples of edited images using a selected basis vector. The latent basis vector could be conditioned on prompts and it facilitates text-aligned manipulations.

metric of the corresponding $\mathcal{H}$. In other words, the norm of $\mathbf{v} \in \mathcal{T}_{\mathbf{x}}$ is measured by the norm of corresponding codomain tangent vector:

$$||\mathbf{v}||^2_{\text{pb}} \triangleq \langle \mathbf{u}, \mathbf{u} \rangle_{\mathbf{h}} = \mathbf{v}^\top J_{\mathbf{x}}^\top J_{\mathbf{x}} \mathbf{v}. \tag{1}$$

## 3.2  Finding local latent basis

Using the pullback metric, we define the local latent vector $\mathbf{v} \in \mathcal{T}_{\mathbf{x}}$ that shows a large variability in $\mathcal{T}_{\mathbf{h}}$. We find a unit vector $\mathbf{v}_1$ that maximizes $||\mathbf{v}||^2_{\text{pb}}$. By maximizing $||\mathbf{v}||^2_{\text{pb}}$ while remaining orthogonal to $\mathbf{v}_1$, one can obtain the second unit vector $\mathbf{v}_2$. This process can be repeated to have $n$ latent directions of $\{\mathbf{v}_1, \mathbf{v}_2, \cdots, \mathbf{v}_n\}$ in $\mathcal{T}_{\mathbf{x}}$. In practice, $\mathbf{v}_i$ corresponds to the $i$-th right singular vector from the singular value decomposition (SVD) of $J_{\mathbf{x}} = U\Lambda V^\top$, i.e., $J_{\mathbf{x}} \mathbf{v}_i = \Lambda_i \mathbf{u}_i$. Since the Jacobian of too many parameters is not tractable, we use a *power method* [18, 34, 19] to approximate the SVD of $J_{\mathbf{x}}$ (See Appendix D for the time complexity and Appendix F for the detailed algorithm).

Henceforth, we refer to $\mathcal{T}_{\mathbf{x}}$ as a local latent subspace, and $\{\mathbf{v}_1, \mathbf{v}_2, \cdots, \mathbf{v}_n\}$ as the corresponding local latent basis.

$$\mathcal{T}_{\mathbf{x}} \triangleq \text{span}\{\mathbf{v}_1, \mathbf{v}_2 \cdots, \mathbf{v}_n\}, \text{ where } \mathbf{v}_i \text{ is } i\text{-th right singular vector of } J_{\mathbf{x}}. \tag{2}$$

Using the linear transformation between $\mathcal{T}_{\mathbf{x}}$ and $\mathcal{T}_{\mathbf{h}}$ via the Jacobian $J_{\mathbf{x}}$, one can also obtain corresponding directions in $\mathcal{T}_{\mathbf{h}}$. In practice, $\mathbf{u}_i$ corresponds to the $i$-th left singular vector from the SVD of $J_{\mathbf{x}}$. After selecting the top $n$ (e.g., $n = 50$) directions of large eigenvalues, we can approximate any vector in $\mathcal{T}_{\mathbf{h}}$ with a finite basis, $\{\mathbf{u}_1, \mathbf{u}_2, \cdots, \mathbf{u}_n\}$. When we refer to a local tangent space henceforth, it means the $n$-dimensional low-rank approximation of the original tangent space.

$$\mathcal{T}_{\mathbf{h}} \triangleq \text{span}\{\mathbf{u}_1, \mathbf{u}_2 \cdots, \mathbf{u}_n\}, \text{ where } \mathbf{u}_i \text{ is the } i\text{-th left singular vector of } J_{\mathbf{x}}. \tag{3}$$

The collection of local latent basis vectors, $\{\mathbf{v}_1, \mathbf{v}_2, \cdots, \mathbf{v}_n\}$, obtained through our proposed method, can be interpreted as a *signal* that the model is highly response to for a given $\mathbf{x}$. On the other hand, the basis of the local tangent space, denoted as $\{\mathbf{u}_1, \mathbf{u}_2 \cdots, \mathbf{u}_n\}$, can be viewed as the corresponding *representation* associated with the signal.

In Stable Diffusion, the prompt also influences the Jacobian, which means that the local basis also depends on it. We can utilize any prompt to obtain a local latent basis, and different prompts create distinct geometrical structures. For the sake of brevity, we will omit the word *local* unless it leads to confusion.

## 3.3  Generating edited images with $\mathbf{x}$-space guidance

A naïve approach for manipulating a latent variable $\mathbf{x}$ using a latent vector $\mathbf{v}$ is through simple addition, specifically $\mathbf{x} + \gamma\mathbf{v}$. However, using the naïve approach sometime leads to noisy image

generation. To address this issue, instead of directly using the basis for manipulation, we use a basis vector that has passed through the decoder once for manipulation. The **x**-space guidance is defined as follows

$$\tilde{\mathbf{x}}_{\text{XG}} = \mathbf{x} + \gamma[\epsilon_\theta(\mathbf{x} + \mathbf{v}) - \epsilon_\theta(\mathbf{x})] \tag{4}$$

where $\gamma$ is a hyper-parameter controlling the strength of editing and $\epsilon_\theta$ is a diffusion model. Equation 4 is inspired by classifier-free guidance, but the key difference is that it is directly applied in the latent space $\mathcal{X}$. Our **x**-space guidance provides qualitatively similar results to direct addition, while it shows better fidelity. (See Appendix C for ablation study.)

### 3.4 The overall process of image editing

In this section, we summarize the entire editing process with five steps: 1) The input image is inverted into initial noise $\mathbf{x}_T$ using DDIM inversion. 2) $\mathbf{x}_T$ is gradually denoised until $t$ through DDIM generation. 3) Identify local latent basis $\{\mathbf{v}_1, \cdots, \mathbf{v}_n\}$ using the pullback metric at $t$. 4) Manipulate $\mathbf{x}_t$ along the one of basis vectors using the **x**-space guidance. 5) The DDIM generation is then completed with the modified latent variable $\tilde{\mathbf{x}}_t$. Figure 2 illustrates the entire editing process.

In the context of a text-to-image model, such as Stable Diffusion, it becomes possible to include textual conditions while deriving local basis vectors. Although we do not use any text guidance during DDIM inversion and generation, a local basis with text conditions enables semantic editing that matches the given text. Comprehensive experiments can be found in Section 4.1.

It is noteworthy that our approach needs no extra training and simplifies image editing by only adjusting the latent variable within a single timestep.

### 3.5 Editing various samples with parallel transport

Let us consider a scenario where our aim is to edit ten images, changing straight hair to curly hair. Due to the nature of the unsupervised image editing method, it is becomes imperative to manually check the semantic relevance of the latent basis vector in the edited results. Thus, to edit every samples, we have to manually find a straight-to-curly basis vector for individual samples.

One way to alleviate this tedious task is to apply the curly hair vector obtained from one image to other images. However, the basis vector $\mathbf{v} \in \mathcal{T}_\mathbf{x}$ obtained at $\mathbf{x}$ cannot be used for the other sample $\mathbf{x}'$ because $\mathbf{v} \notin \mathcal{T}_{\mathbf{x}'}$. Thus, in order to apply the direction we obtained to another sample, it is necessary to relocate the extracted direction to a new tangent space.

To achieve this, we use parallel transport that moves $\mathbf{v}_i$ onto the new tangent space $\mathcal{T}_{\mathbf{x}'}$. In the realm of differential geometry, parallel transport is a technique to relocate a tangent vector to different position with minimal direction change, while keeping the vector tangent on the manifold [48]. It is notable that in curved space, parallel transport can significantly modify the original vector. Therefore, it is beneficial to apply the parallel transport in $\mathcal{H}$, where relatively flatter than $\mathcal{X}$.

The process of moving a tangent vector $\mathbf{v} \in \mathcal{T}_\mathbf{x}$ to $\mathbf{v}' \in \mathcal{T}_{\mathbf{x}'}$ through parallel transport in $\mathcal{H}$ is summarized as follows. First, we convert the latent direction $\mathbf{v}_i \in \mathcal{T}_\mathbf{x}$ to the corresponding direction of $\mathbf{u}_i \in \mathcal{T}_\mathbf{h}$. Second, we apply the parallel transport $\mathbf{u}_i \in \mathcal{T}_\mathbf{h}$ to $\mathbf{u}'_i \in \mathcal{T}_{\mathbf{h}'}$, where $\mathbf{h}' = f(\mathbf{x}')$. In the general case, parallel transport involves iterative projection and normalization on the tangent space along the path connecting two points [48]. However, in our case, we assume that $\mathcal{H}$ has Euclidean geometry. Therefore, we move $\mathbf{u}$ directly onto $\mathcal{T}_{\mathbf{h}'}$ through projection, without the need for an iterative process. Finally, transform $\mathbf{u}'_i$ into $\mathbf{v}'_i \in \mathcal{X}$. Using this parallel transport of $\mathbf{v}_i \rightarrow \mathbf{v}'_i$ via $\mathcal{H}$, we can apply the local latent basis obtained from $\mathbf{x}$ to edit or modify the input $\mathbf{x}'$.

## 4 Findings and results

In this section, we analyze the geometric structure of DMs with our method. § 4.1 demonstrates that the latent basis found by our method can be used for image editing. In § 4.2, we investigate how the geometric structure of DMs evolves as the generative process progresses. Lastly, in § 4.3, we examine how the geometric properties of the text-condition model change with a given text.

The implementation details of our work are provided on Appendix B. The source code for our experiments can be found at https://github.com/enkeejunior1/Diffusion-Pullback.

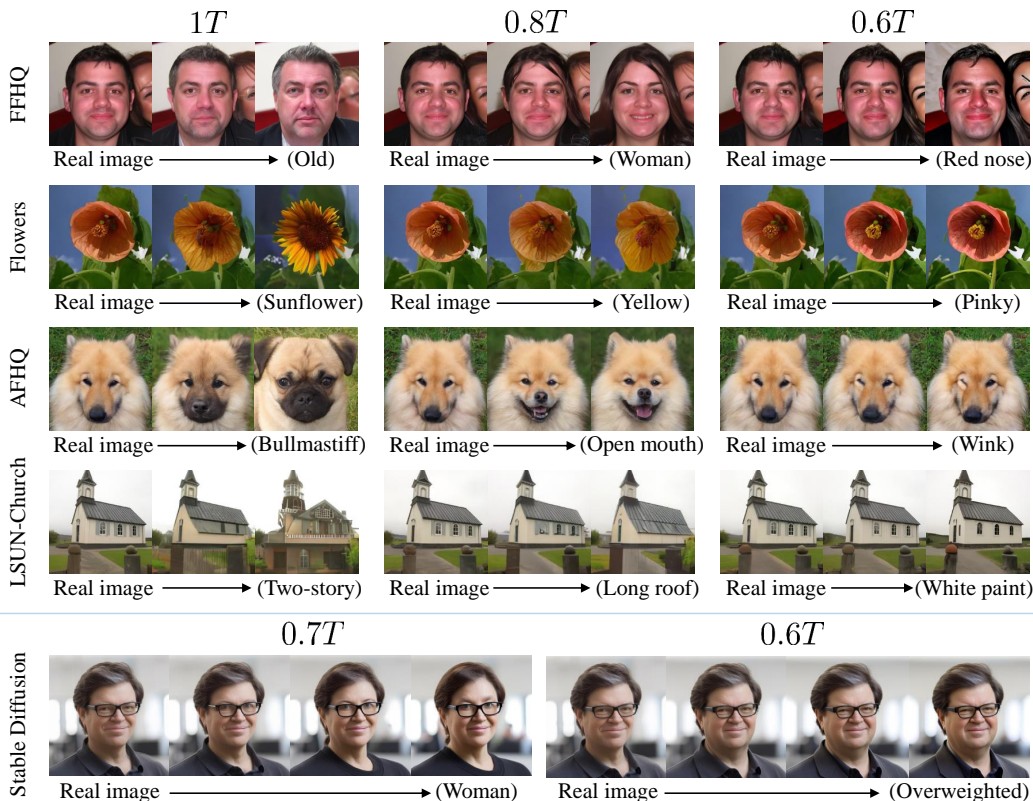

Figure 3: **Examples of image edition using the latent basis.** The attributes are manually interpreted since the editing directions are not intentionally supervised. For Stable Diffusion, we used an empty string as a prompt. Each column represents edits made at different diffusion timesteps ($0.6T$, $0.8T$, and $T$ for the unconditional diffusion model; $0.6T$ and $0.7T$ for Stable Diffusion).

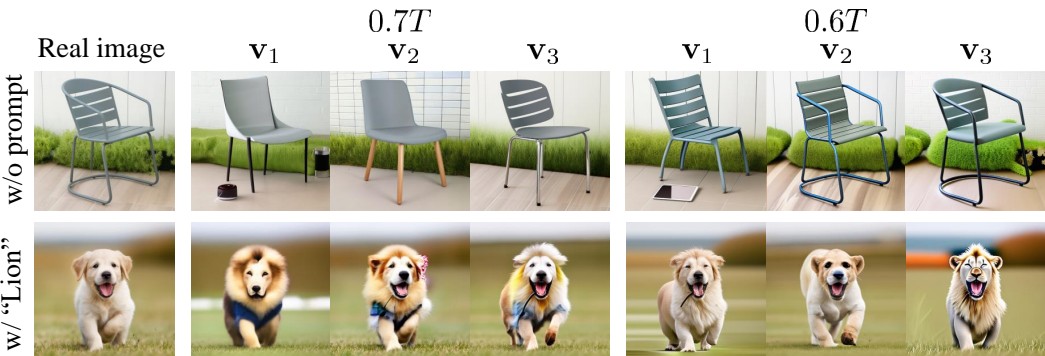

Figure 4: **Examples of image edition using top-3 latent basis vectors.** Each column is edited using a different latent vector $\{\mathbf{v}_1, \mathbf{v}_2, \mathbf{v}_3\}$. Each group of columns represents edits made at different diffusion timesteps ($0.6T$ and $0.7T$). Notably, when given the "Lion" prompt, it is evident that all the top latent basis vectors align with the direction of the prompt.

## 4.1 Image editing with the latent basis

In this subsection, we demonstrate the capability of our discovered latent basis for image editing. To extract the latent variables from real images for editing purposes, we use DDIM inversion. In experiments with Stable Diffusion (SD), we do not use guidance, i.e., unconditional sampling, for both DDIM inversion and DDIM sampling processes. This ensures that our editing results solely depend on the latent variable, not on other factors such as prompt conditions.

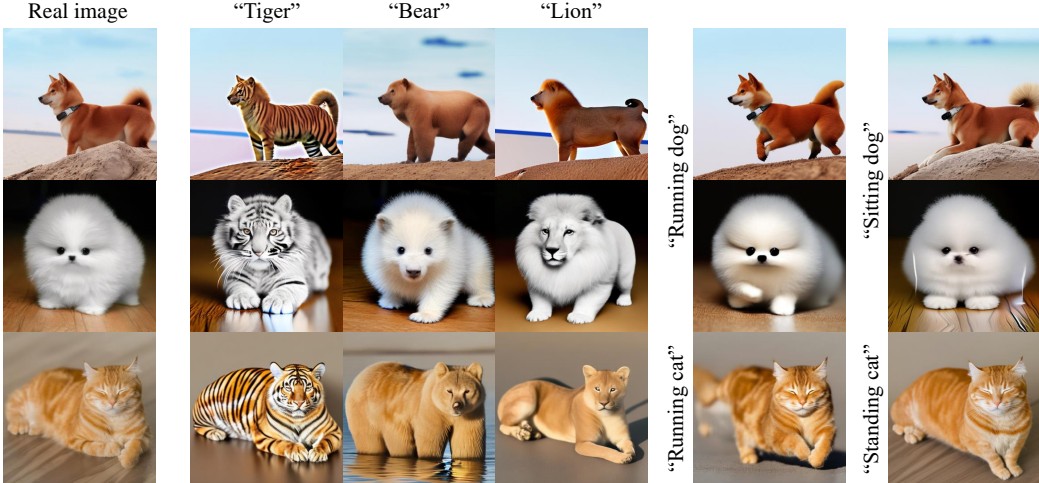

Figure 5: **Examples of image edition using latent basis vectors discovered with various prompts.** Each column is edited using the latent basis vector obtained from a different text prompt. Importantly, our method employs each prompt only once to derive the local latent basis.

Figures 2 and 3 illustrate the example results edited by the latent basis found by our method. The latent basis clearly contains semantics such as age, gender, species, structure, and texture. Note that editing at timestep $T$ yields coarse changes such as age and species. On the other hand, editing at timestep $0.6T$ leads to fine changes, such as nose color and facial expression.

Figure 4 demonstrates the example results edited by the various latent basis vectors. Interestingly, using the text "lion" as a condition, the entire latent basis captures lion-related attributes. Furthermore, Figure 5 shows that the latent basis aligns with the text not only in terms of object types but also in relation to pose or action. For a qualitative comparison with other state-of-the-art image editing methods, refer to Appendix D. For more examples of editing results, refer to Appendix G.

## 4.2 Evolution of latent structures during generative processes

In this subsection, we demonstrate how the latent structure evolves during the generative process and identify three trends. 1) The frequency domain of the latent basis changes from low to high frequency. It agrees with the previous observation that DMs generate samples in coarse-to-fine manner. 2) The difference between the tangent spaces of different samples increases over the generative process. It implies finding generally applicable editing direction in latent space becomes harder in later timesteps. 3) The differences of tangent spaces between timesteps depend on the complexity of the dataset.

**Latent bases gradually evolve from low- to high-frequency structures.** Figure 6 is the power spectral density (PSD) of the discovered latent basis over various timesteps. The early timesteps contain a larger portion of low frequency than the later timesteps and the later timesteps contain a larger portion of high frequency.

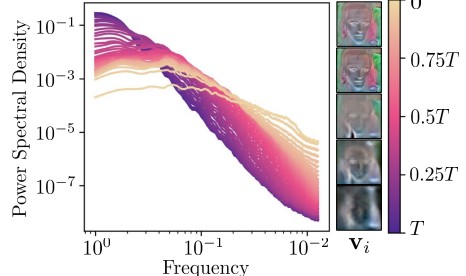

Figure 6: **Power Spectral Density (PSD) of latent basis.** The PSD at $t = T$ (purple) exhibits a greater proportion of low-frequency signals, while the PSD at smaller $t$ (beige) reveals a larger proportion of high-frequency signals. The latent vectors $\mathbf{v}_i$ are min-max normalized for visual clarity.

This suggests that the model focuses on low-frequency signals at the beginning of the generative process and then shifts its focus to high-frequency signals over time. This result strengthens the common understanding about the coarse-to-fine behavior of DMs over the generative process [12, 15].

**The discrepancy of tangent spaces from different samples increases along the generative process.** To investigate the geometry of the tangent basis, we employ a metric on the Grassmannian manifold.

The Grassmannian manifold is a manifold where each point is a vector space, and the metric defined above represents the distortion across various vector spaces. We use *geodesic metric* [10, 56] to define the discrepancy between two subspaces $\{\mathcal{T}^{(1)}, \mathcal{T}^{(2)}\}$:

$$D_{\text{geo}}(\mathcal{T}^{(1)}, \mathcal{T}^{(2)}) = \sqrt{\sum_k \theta_k^2}, \tag{5}$$

where $\theta_k$ denotes the $k$-th principle angle between $\mathcal{T}^{(1)}$ and $\mathcal{T}^{(2)}$. Intuitively, the concept of geodesic metric can be understood as an angle between two vector spaces. Here, the comparison between two different spaces was conducted for $\{\mathcal{T}_{\mathbf{h}_1}, \mathcal{T}_{\mathbf{h}_2}\}$. Unlike the $\mathcal{X}$, the $\mathcal{H}$ assumes a Euclidean space which makes the computation of geodesic metric that requires an inner product between tangent spaces easier. The relationship between tangent space and latent subspace is covered in more detail in Appendix E.

Figure 7 demonstrates that the tangent spaces of the different samples are the most similar at $t = T$ and diverge as timestep becomes zero.

Moreover, the similarity across tangent spaces allows us to effectively transfer the latent basis from one sample to another through parallel transport as shown in Figure 8. In $T$, where the tangent spaces are homogeneous, we consistently obtain semantically aligned editing results. On the other hand, parallel transport at $t = 0.6T$ does not lead to satisfactory editing because the tangent spaces are hardly homogeneous. Thus, we should examine the similarity of local subspaces to ensure consistent editing across samples.

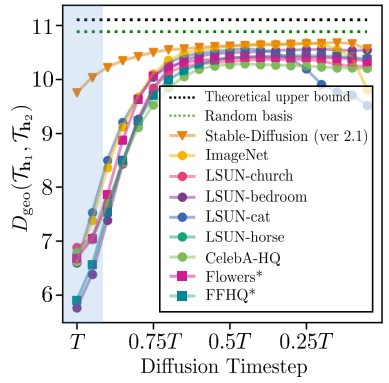

Figure 7: **Geodesic distance across tangent space of different samples at various diffusion timesteps.** Each point represents the average geodesic distance between pairs of 15 samples. It is notable that the similarity of tangent spaces among different samples diminishes as the generative process progresses.

**DMs trained on simpler datasets exhibit more consistent tangent spaces over time.** In Figure 9 (a), we provide a distance matrix of the tangent spaces across different timesteps, measured by the geodesic metric. We observe that the tangent spaces are more similar to each other when a model is trained on CelebA-HQ, compared to ImageNet. To verify this trend, we measure the geodesic distances between tangent spaces of different timesteps and plot the average distances of the same

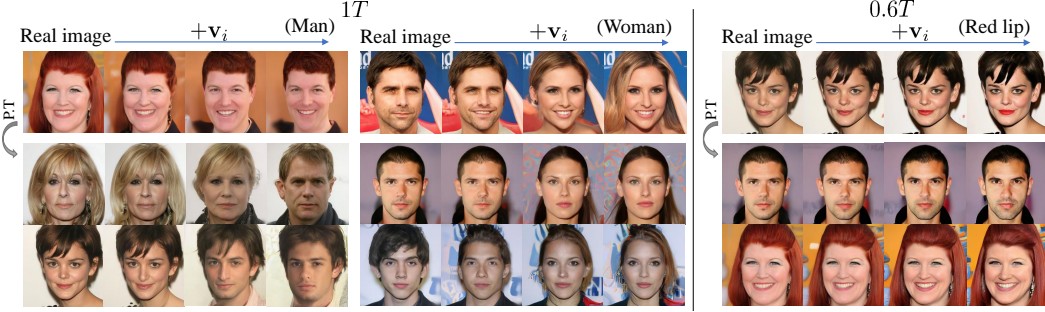

Figure 8: **Examples of image edition using parallel transport.** The first row demonstrates the results of editing with their respective latent vectors, while the subsequent rows exhibit the results of editing through the parallel transport (P.T) of the latent vectors used in the first row. The latent vector performs effectively when $t = T$ (left and middle), but comparatively less satisfactorily for $0.6T$ (right).

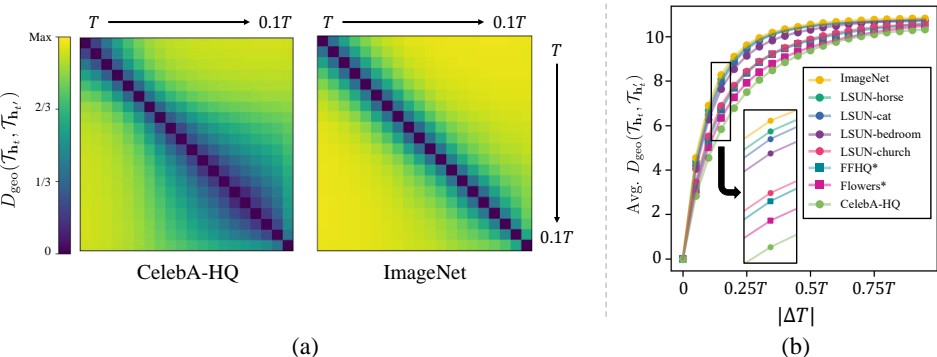

(a)                                                                            (b)

Figure 9: **Simpler datasets lead to more similar tangent spaces across diffusion timesteps.** (a) Distance matrix visualization of tangent space measured by geodesic metric across various timesteps. (b) Average geodesic distance based on timestep differences, indicating that the complexity of the dataset correlates with greater distances between tangent spaces.

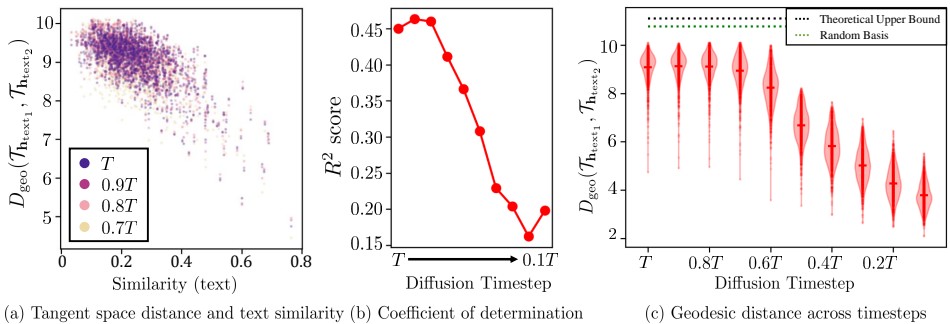

(a) Tangent space distance and text similarity (b) Coefficient of determination    (c) Geodesic distance across timesteps

Figure 10: **Similar prompts create similar tangent spaces, and the impact of the prompt decreases as the generative process progresses.** (a) The horizontal axis represents the CLIP similarity between two different prompts, while the vertical axis represents the geodesic distance in the tangent space from each prompt. Different colors represent various diffusion timesteps. A negative relationship is observed between prompt similarity and tangent space distance. (b) The $R^2$ score of the linear regression between clip similarity and geodesic distance of tangent spaces decreases throughout the generative process. (c) Each point represents the distance between tangent spaces created from different prompts. Until around $t = 0.7T$, the distance between tangent spaces is very large, but it gradually decreases thereafter. This indicates that the influence of the prompt on the tangent space diminishes.

difference in timestep in Figure 9 (b). As expected, we find that DMs trained on datasets, that are generally considered simpler, have similar local tangent spaces over time.

## 4.3    Effect of conditioning prompts on the latent structure

In this subsection, we aim to investigate how prompts influence the generative process from a geometrical perspective. We randomly sampled 50 captions from the MS-COCO dataset [29] and used them as text conditions.

**Similar text conditions induce similar tangent spaces.**    In Figure 10 (a), we observe a negative correlation between the CLIP similarity of texts and the distance between tangent spaces. In other words, when provided with similar texts, the tangent spaces are more similar. The linear relationship between the text and the discrepancy of the tangent spaces is particularly strong in the early phase of the generative process as shown by $R^2$ score in Figure 10 (b).

**The generative process depends less on text conditions in later timesteps.**    Figure 10 (c) illustrates the distances between local tangent spaces for given different prompts with respect to the timesteps.

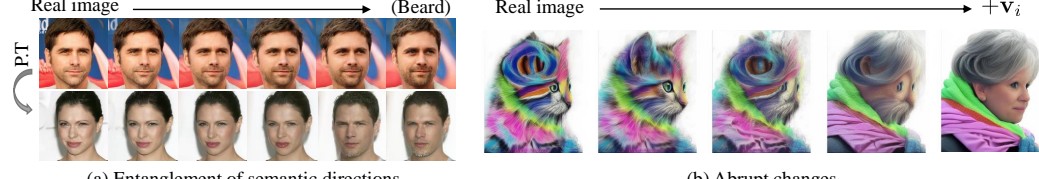

(a) Entanglement of semantic directions  (b) Abrupt changes

Figure 11: **Limitations.** (a) Entanglement between attributes due to dataset biases (b) Abrupt changes in Stable Diffusion.

Notably, as the diffusion timestep approaches values below $0.7T$, the distances between the local tangent spaces start to decrease. It implies that the variation due to walking along the local tangent basis depends less on the text conditions, i.e., the text less influences the generative process, in later timesteps. It is a possible reason why the correlation between the similarity of prompts and the similarity of tangent spaces reduces over timesteps.

## 5 Discussion

In this section, we provide additional intuitions and implications. It is interesting that our latent basis usually conveys disentangled attributes even though we do not adopt attribute annotation to enforce disentanglement. We suppose that decomposing the Jacobian of the encoder in the U-Nets naturally yields disentanglement to some extent. However, it does not guarantee the perfect disentanglement and some directions are entangled. For example, the editing for beard converts a female subject to a male as shown in Figure 11 (a). This kind of entanglement often occurs in other editing methods due to the dataset bias: female faces seldom have beard.

While our method has shown effectiveness in Stable Diffusion, more research is needed to fully validate its potential. We have observed that some of the discovered latent vector occasionally leads to abrupt changes during the editing process in Stable Diffusion, as depicted in Figure 11 (b). This observation highlights the complex geometry of $\mathcal{X}$ in achieving seamless editing. Exploring this topic in future research is an interesting area to delve into.

Our approach is broadly applicable when the feature space in the DM adheres to a Euclidean metric, as demonstrated by $\mathcal{H}$. This characteristic has been observed in the context of U-Net within Kwon et al. [26]. It would be intriguing to investigate if other architectural designs, especially those similar to transformer structures as introduced in [42, 53], also exhibit a Euclidean metric.

Despite these limitations, our method provides a significant advance in the field of image editing for DMs, and provides a deep understanding of DM through several experiments.

## 6 Conclusion

We have analyzed the latent space of DMs from a geometrical perspective. We used the pullback metric to identify the latent and tangent bases in $\mathcal{X}$ and $\mathcal{H}$. The latent basis found by the pullback metric allows editing images by traversal along the basis. We have observed properties of the bases in two aspects. First, we discovered that 1) the latent bases evolve from low- to high-frequency components; 2) the discrepancy of tangent spaces from different samples increases along the generative process; and 3) DMs trained on simpler datasets exhibit more consistent tangent spaces over timesteps. Second, we investigated how the latent structure changes based on the text conditions in Stable Diffusion, and discovered that similar prompts make tangent space analogous but its effect becomes weaker over timesteps. We believe that a better understanding of the geometry of DMs will open up new possibilities for adopting DMs in useful applications.

## 7 Acknowledgement

This work was supported in part by the Creative-Pioneering Researchers Program through Seoul National University, the National Research Foundation of Korea (NRF) grant (Grant No. 2022R1A2C1006871) (J. J.), KIAS Individual Grant [AP087501] via the Center for AI and Natural Sciences at Korea Institute for Advanced Study, and the National Research Foundation of Korea (NRF) grant (RS-2023-00223062).

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
