Table A1: Hyper-parameter settings.

| model | $t_{edit}$ | inversion step | $\gamma$ | $n$ | $t_{boost}$ |
|---|---|---|---|---|---|
| Stable Diffusion | $0.7T$ | 100 | 1 | 50 | $\times$ |
| | $0.6T$ | 100 | 2 | 50 | $\times$ |
| Unconditional DMs | $T$ | 100 | 0.5 | 50 | $0.2T$ |
| | $0.8T$ | 100 | 1 | 50 | $0.2T$ |
| | $0.6T$ | 100 | 4 | 50 | $0.2T$ |

## A    Societal Impacts & Ethics Statements

Our research endeavors to unravel the geometric structures of the diffusion model and facilitate high-quality image editing within its framework. While our primary application resides within the creative realm, it is important to acknowledge that image manipulation techniques, such as the one proposed in our method, hold the potential for misuse, including the dissemination of misinformation or potential privacy implications. Therefore, the continuous advancement of technologies aimed at thwarting or identifying manipulations rooted in generative models remains of utmost significance.

## B    Implementation details

**Models and datasets**    We validate our method and provide analyses on various models using the official code and pre-trained checkpoints. The available combinations of the models and the datasets are: DDPM [22] on ImageNet [16], LSUN-church/bedroom/cat/horse [58], and CelebA-HQ [23]; and DDPM trained with *P2 weighting* [12] on FFHQ [24], Flowers [58] and AFHQ [13]. We also use Stable Diffusion (SD) version 2.1 [46] for the text-conditional diffusion model.

For image editing, we use the official codes and pre-trained checkpoints for all baselines and keep the parameters *frozen*. For analysis, we compare models with the same diffusion scheduling (linear schedule) and resolutions ($256^2$) to ensure a fair comparison, except Stable Diffusion.

Table B1 summarizes various hyperparameter settings in our experiments. Specific details not covered in the main text are discussed in the following paragraphs.

**Edit timestep** ($t_{edit}$)    For unconditional DMs, we show the editing results at $t_{edit} \in \{T, 0.8T, 0.6T\}$, while for Stable Diffusion, we show the editing results at $t_{edit} \in \{0.7T, 0.6T\}$. Note that our method allows manipulation at any timestep.

**Inversion step**    We conduct real image editing with DDIM inversion [51]. We set the number of steps to 100 for obtaining the latent variable $\mathbf{x}_T$ and all experiments.

**x-space guidance scale** ($\gamma$)    The value of $\gamma$ determines the magnitude of a single editing step by x-space guidance. Fortunately, through experimentation, we observed that the value of $\gamma$ does not have a significant impact on image quality unless it is excessively large.

**Low-rank approximation** ($n$)    We employ a low-rank approximation of the tangent space using $n = 50$ for all settings.

**Quality boosting** ($t_{boost}$)    While DDIM alone already generates high-quality images, Karras et al. [25] showed that including stochasticity in the process improves image quality and Kwon et al. [26] suggest similar technique: adding stochasticity at the end of the generative process. We employ this technique in our experiments on every experiment after $t = 0.2T$, except Stable Diffusion.

**Computing resource**    For power-method approximation with $n = 50$, it spends about 3-4 minutes on a single NVIDIA RTX 3090 (24GB). As $n$ specifies the number of bases, it can be as small as a user want to use for image editing. Reducing $n$ provides faster runtime, e.g., 10 seconds for $n = 3$.

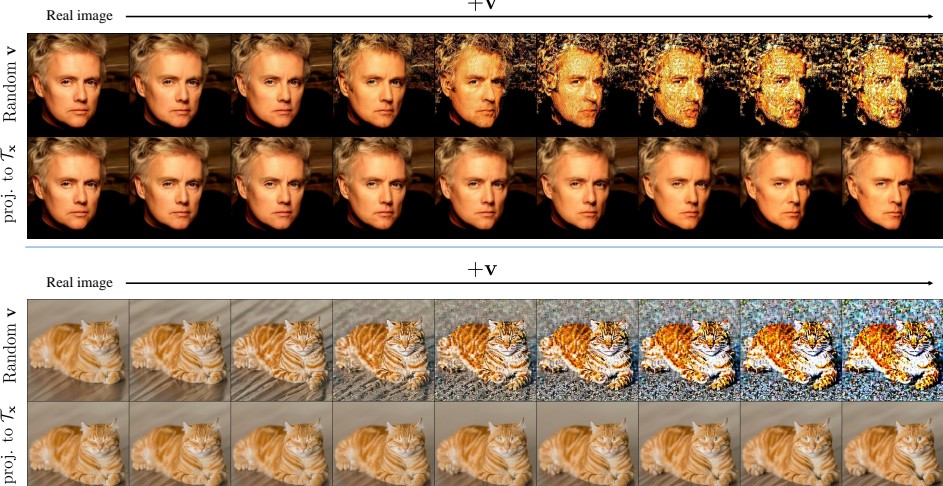

Figure A1: **Importance of the discovered latent directions.** Random direction experiments with CelebA-HQ pre-trained model (top) and Stable Diffusion (bottom). Adding random directions instead of latent directions severely distorts the resulting images. When we perform edits along the projection onto the latent subspace $\mathcal{T}_\mathbf{x}$, the generated image presents a semantically meaningful transformation.

## C  Ablation study

In this section, we validate our method with ablation study.

**Random $\mathbf{v}$**  To demonstrate the meaningfulness of the latent basis found by our method, we qualitatively compare its effect to naïve baseline: random directions. The first row in Figure A1 shows that manipulating the images with a random vector '$\mathbf{v}$' does not result in semantic editing but rather degrades images. The second row shows the results of projecting the random '$\mathbf{v}$' onto our obtained latent subspace. The projected results exhibit semantic manipulation such as pose changes without image distortion. It indicates that the found latent subspace captures semantics in the latent space effectively.

**$\mathbf{x}$-space guidance**  Figure A2 demonstrates the the effectiveness of $\mathbf{x}$-space guidance compared to a straightforward alternative: simple addition. First, $\mathbf{x}$-space guidance produces higher quality images with similar meaning. Especially Stable Diffusion apparently benefits from $\mathbf{x}$-space guidance regarding smoothness of the editing strength and artifacts. The difference is more significant at $t = 0.6T$. Note that the meaning of the same directions may slightly differ between the two settings due to non-linearity of the U-Net.

Currently, we do not have a deeper understanding of the underlying principles of $\mathbf{x}$-space guidance. Exploring the reasons behind its ability to improve manipulation quality would be an interesting direction for future work.

## D  Comparative experiment to other state-of-the-art (SoTA) editing methods

We conduct qualitative comparisons with text-guided image editing methods. Our SoTA baseline methods include: (i) SDEdit [33], (ii) Pix2Pix-zero [39], (iii) PnP [54], and (iv) Instruct Pix2Pix [7]. All comparisons were performed using the official code. Please refer to Figure A3 for the qualitative results.

We also compare the time complexity of each method. For a fair comparison, we only identify the first singular vector $\mathbf{v}_1$, i.e., $n = 1$, and set the number of DDIM steps to 50. All experiments were conducted on an Nvidia RTX 3090. The runtime for each method is summarized in Table A2.

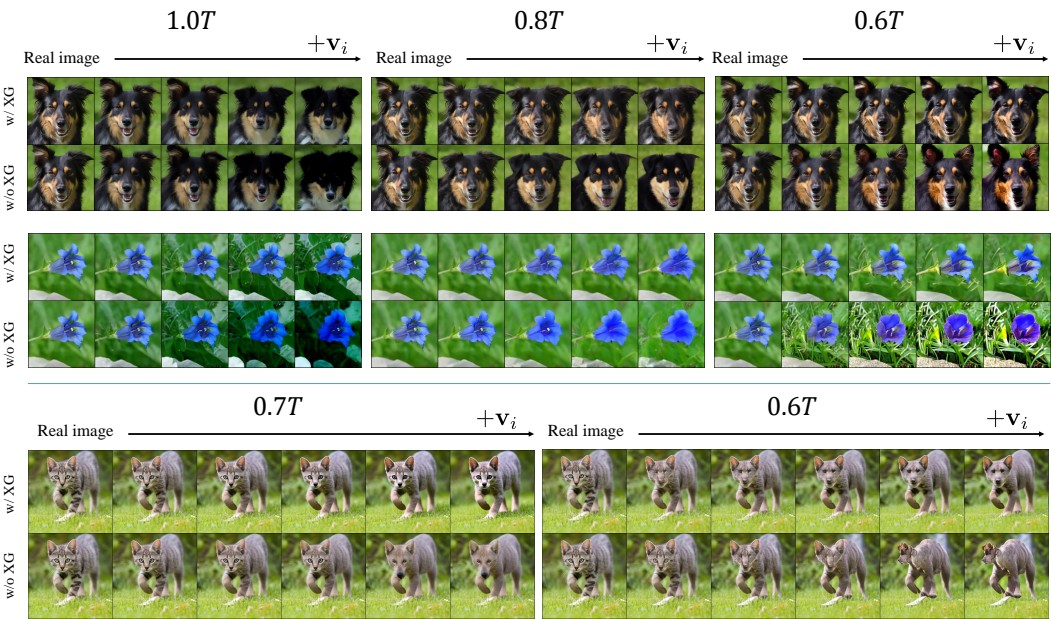

Figure A2: **Importance of the x-space guidance.** x-space guidance experiments with AFHQ (top), Flowers pre-trained model (middle), and Stable Diffusion (bottom). x-space guidance helps achieve qualitatively similar editing while preserving the content of the original image.

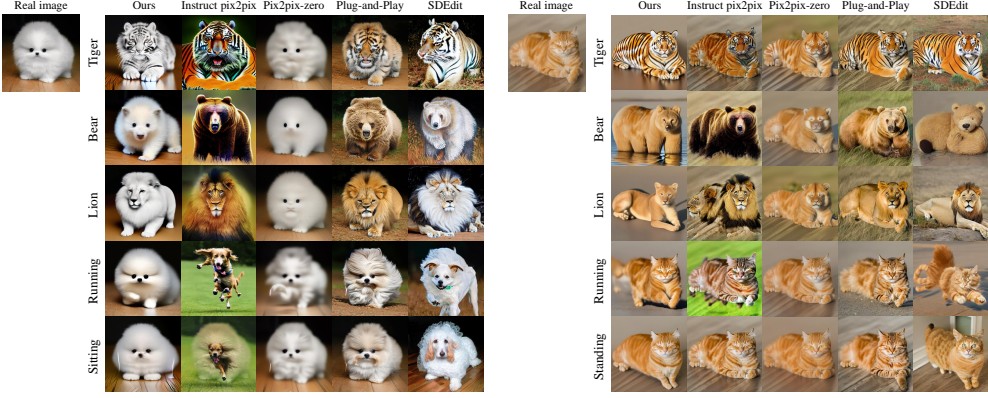

Figure A3: **Comparison with various image editing methods.** Our approach empowers image editing that aligns seamlessly with text conditions while upholding object identity. In contrast, alternative methods exhibit deficiencies such as: inadequate preservation of object structure (Instruct pix2pix), inefficacious manipulation (Pix2pix-zero), or challenges in maintaining identity fidelity (Plug-and-Play, SDEdit).

The computation cost of our method remains comparable to other approaches, although the Jacobian approximation takes around 2.5 seconds for $n = 1$. This is because we only need to identify the latent basis vector once at a specific timestep. Furthermore, our approach does not require additional preprocessing steps like generating 100 prompts with GPT and obtaining embedding vectors (as in Pix2Pix-zero), or storing feature vectors, queries, and key values (as in PnP). Our method also does not require finetuning (as in Instruct Pix2Pix). This leads to a significantly reduced total editing process time in comparison to other methods.

Table A2: **Comparisons of the time complexity of state-of-the-art editing methods**

| Image Edit Method | Running time | Preprocessing |
|---|---|---|
| Ours | 11 sec | N/A |
| SDEdit | 4 sec | N/A |
| Pix2Pix-zero | 25 sec | 4 min |
| PnP | 10 sec | 40 sec |
| Instruct Pix2Pix | 11 sec | N/A |

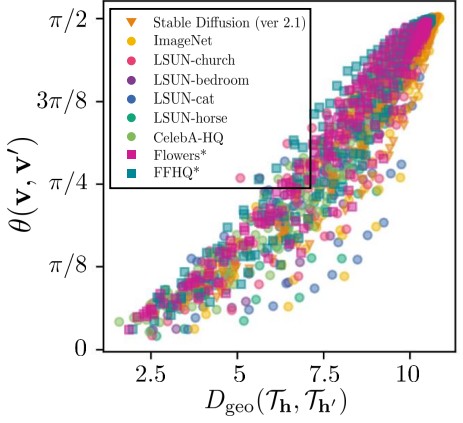

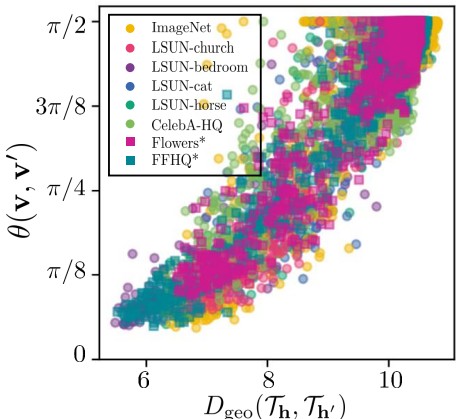

(a) Parallel transport across timesteps

(b) Parallel transport across samples

Figure A4: **Parallel transport between similar tangent spaces creates similar latent directions.** The horizontal axis represents the geodesic distance between tangent spaces from (a) different timesteps (b) different samples at $t \in \{T, 0.9T, \cdots, 0.1T\}$. The vertical axis represents the angle between the original latent direction and transported latent direction. Different colors represent various datasets. A positive relationship is observed between tangent space distance and the distortion induced by parallel transport.

# E    More Discussions

**Why do we measure the geodesic distance of the tangent spaces instead of the latent subspaces?** The geodesic distance on the Grassmannian manifold between two subspaces is defined as the $l_2$-norm of principal angles. To define angles between different vector spaces, an inner product needs to be defined. In our work, we define the inner product in $\mathcal{T}_{\mathbf{x}}$ using the pullback metric. The issue is that the pullback metric is locally defined for each latent subspace $\mathcal{T}_{\mathbf{x}}$ (Eq. (1)). Therefore, measuring angles between distant latent subspaces becomes challenging. On the other hand, $\mathcal{H}$ follows the assumption of the Euclidean metric. Consequently, even for distant tangent spaces, angles can be easily computed using the dot product. In this regard, we measure the similarity between latent subspaces by exploiting the geodesic distance of their corresponding tangent spaces. Furthermore, when compared to $\mathcal{X}$, $\mathcal{H}$ offers the advantage of being a semantic space, making it more suitable for measuring semantic similarity.

**Similar tangent space implies similar latent subspace**    In Figure A4, we calculated the geodesic distance of tangent spaces obtained at different timesteps (or different samples at same the timestep) and the angle between the original latent direction and parallel transported direction between them. It is evident that as the geodesic distance decreases, the amount of distortion during parallel transport also reduces.

Notice that the similarity between tangent spaces implies consistency of latent basis across timesteps. In Figure A5 (b), we parallel transport the latent vector $\mathbf{v}_i$ to various tangent spaces and visualize the outcomes. As expected, when the tangent spaces are similar, the transported vector $\mathbf{v}'_i$ retains the original signal. On the other hand, as we move to more distant timesteps, where the tangent space is farther apart, $\mathbf{v}'_i$ deviates from the original signal.

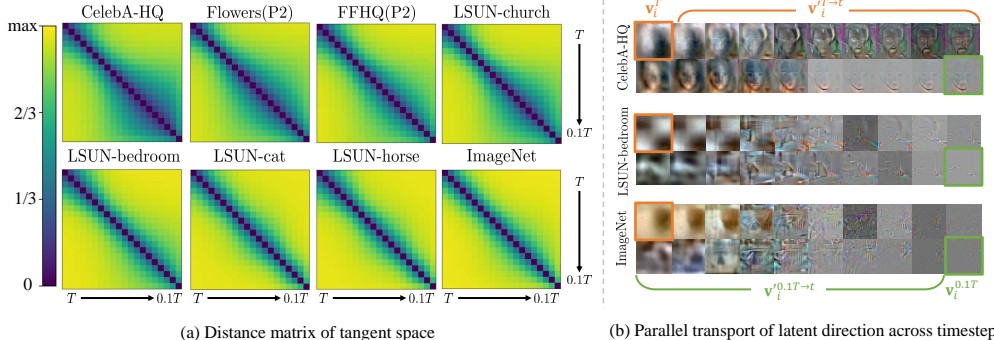

(a) Distance matrix of tangent space      (b) Parallel transport of latent direction across timesteps

Figure A5: **More examples of tangent spaces across diffusion timesteps.** (a) Distance matrix visualization of tangent space measured by geodesic metric across various timesteps. (b) Visualization of the result from parallel transport across timesteps. $\mathbf{v}'^{t_a \to t_b}_i$ denotes the latent vector transported from $t_a$ to $t_b$. Transported vector significantly deviates from the original vector, as the tangent space grows further apart according to the distance matrix. For visualization purposes, $\mathbf{v}_i$ is min-max normalized.

# F    Algorithms

In this section, for reproducibility purposes, we provide the code for two important algorithms. The code is implemented using PyTorch [40].

**Jacobian subspace iteration** The diffusion model has dimensions that are too large in both $\mathcal{X}$ and $\mathcal{H}$, making the computation of the Jacobian infeasible. To overcome this challenge, we attempt the Jacobian subspace iteration algorithm to approximate the singular value of the Jacobian, as proposed in [19]. For details, please refer to Haas et al. [19].

```python
import torch # >= ver 2.0

def local_encoder_pullback(
        x, t, get_h, n=50, chunk_size=25, min_iter=10, max_iter=100,
    convergence_threshold=1e-4,
    ):
    '''
    Args
        - x : tensor ; latent variable
        - t : tensor ; diffusion timestep
        - get_h : function ; return h given x, t
        - n ; low-rank approximation dimension
        - chunk_size ; To avoid OOM error
        - min_iter (max_iter) ; minimum (maximum) number of iteration
        - convergence_threshold ; to check convergence of power-method
    '''
    # set number of chunk to avoid OOM
    num_chunk = n // chunk_size + 1

    # get dimensions of x space and h space
    h_shape = get_h(x, t).shape
    c_i, w_i, h_i = x.size(1), x.size(2), x.size(3)
    c_o, w_o, h_o = h_shape[1], h_shape[2], h_shape[3]

    # power-method
    a = torch.tensor(0., device=x.device, dtype=x.dtype)
    vT = torch.randn(c_i*w_i*h_i, n, device=x.device)
    vT, _ = torch.linalg.qr(vT)
    v = vT.T
    v = v.view(-1, c_i, w_i, h_i)

    for i in range(max_iter):
```

```
32          v = v.to(device=x.device, dtype=x.dtype)
33          v_prev = v.detach().cpu().clone()
34
35          time_s = time.time()
36          u = []
37          v_buffer = list(v.chunk(num_chunk))
38          for vi in v_buffer:
39              g = lambda a : get_h(x + a*vi, t=t)
40              ui = torch.func.jacfwd(
41                  g, argnums=0, has_aux=False, randomness='error'
42              )(a)
43              u.append(ui.detach().cpu().clone())
44          u = torch.cat(u, dim=0)
45          u = u.to(x.device, x.dtype)
46
47          g = lambda x : torch.einsum(
48              'b c w h, i c w h -> b', u, get_h(x, t=t)
49          )
50          v_ = torch.autograd.functional.jacobian(g, x)
51          v_ = v_.view(-1, c_i*w_i*h_i)
52
53          _, s, v = torch.linalg.svd(v_, full_matrices=False)
54          v = v.view(-1, c_i, w_i, h_i)
55          u = u.view(-1, c_o, w_o, h_o)
56
57          convergence = torch.dist(v_prev, v.detach().cpu()).item()
58          if torch.allclose(v_prev, v.detach().cpu(), atol=
       convergence_threshold) and (i > min_iter):
59              break
60
61      # reshape as a x space, h space vector
62      u, s, vT = u.reshape(-1, c_o*w_o*h_o).T.detach(), s.sqrt().detach
       (), v.reshape(-1, c_i*w_i*h_i).detach()
63      return u, s, vT
```

Listing 1: **Jacobian subspace iteration**

**Geodesic metric**    For a detailed discussion on the geodesic metric, please refer to Choi et al. [10] for more information.

```
1  import torch
2
3  def geodesic_metric(U1, U2):
4      _, S, _ = torch.linalg.svd(U1.T @ U2)
5      th = torch.acos(S)
6      return th.norm()
```

Listing 2: **Geodesic metric**

# G  Additional results

## G.1  Latent basis

**Unconditional latent basis**   We provide more examples of image editing using the latent basis. Figure A6, A7, A8, A9 and A10 show that every latent basis produces different results and editing at timestep $T$ yields coarse changes while $0.6T$ leads to fine changes. Stable Diffusion shows a similar trend; $0.7T$ yields coarse changes while $0.6T$ leads to fine changes. The results of $T$ in Stable Diffusion will be covered in the § G.3. Please zoom in for the best view.

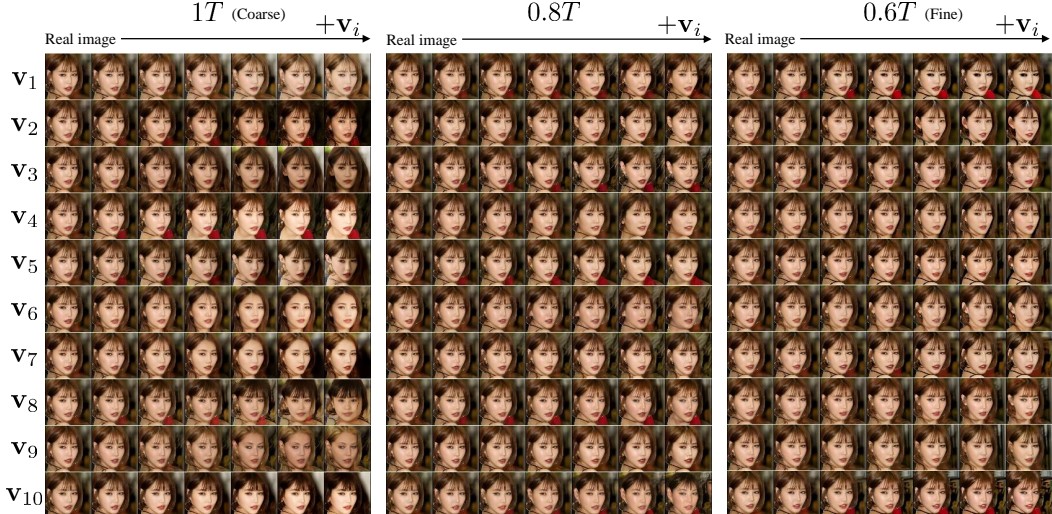

Figure A6: **More examples of image editing using the latent basis in FFHQ.** The editing result using ten $\mathbf{v}_i$' in FFHQ. Each column represents edits made at different diffusion timesteps ($0.6T$, $0.8T$, and $1T$). Editing at timestep $1T$ yields coarse changes. On the other hand, editing at timestep $0.6T$ leads to fine changes. Please zoom in for the best view.

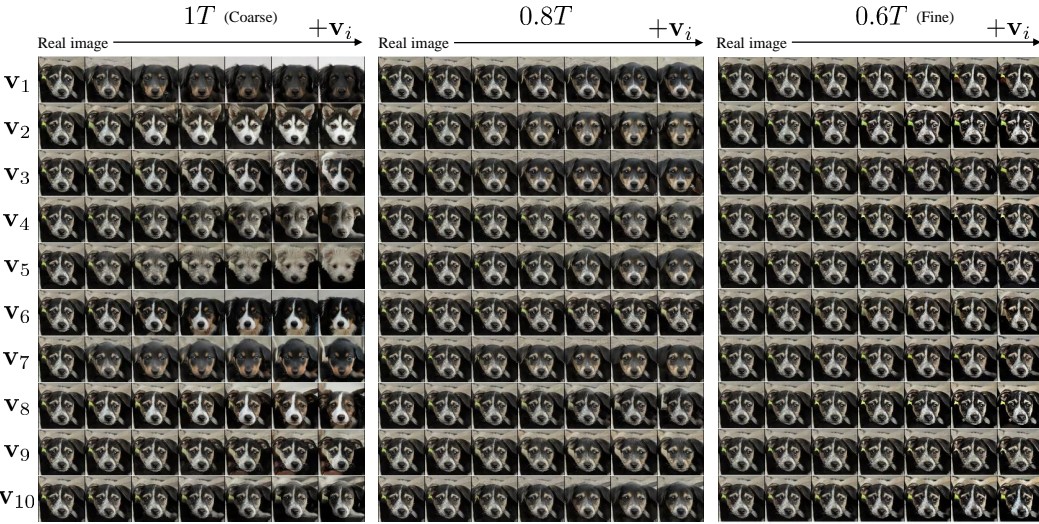

Figure A7: **More examples of image editing using the latent basis in AFHQ.** The editing result using ten $\mathbf{v}_i$' in AFHQ. Each column represents edits made at different diffusion timesteps ($0.6T$, $0.8T$, and $1T$). Editing at timestep $1T$ yields coarse changes. On the other hand, editing at timestep $0.6T$ leads to fine changes. Please zoom in for the best view.

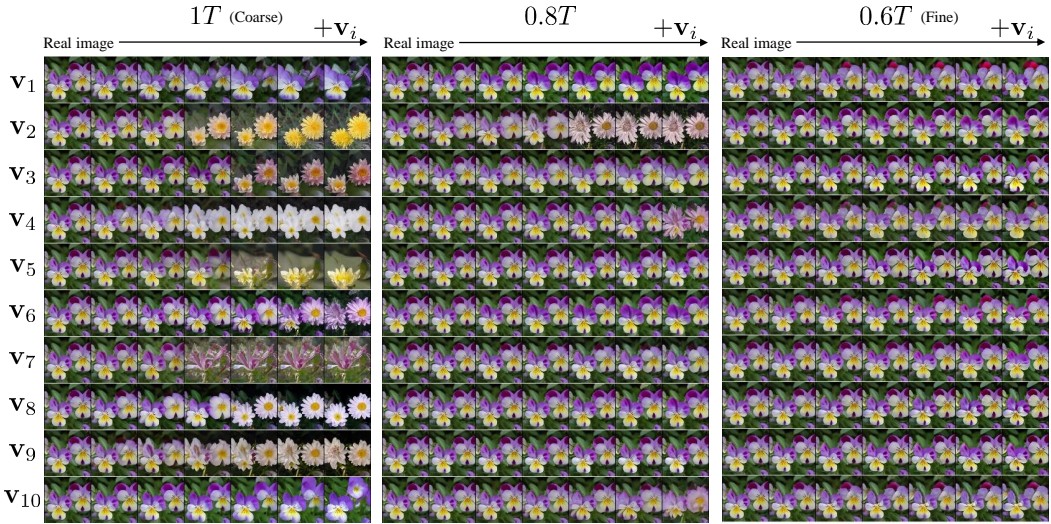

Figure A8: **More examples of image editing using the latent basis in Flowers.** The editing result using ten $\mathbf{v}_i$' in Flowers. Each column represents edits made at different diffusion timesteps ($0.6T$, $0.8T$, and $1T$). Editing at timestep $1T$ yields coarse changes. On the other hand, editing at timestep $0.6T$ leads to fine changes. Please zoom in for the best view.

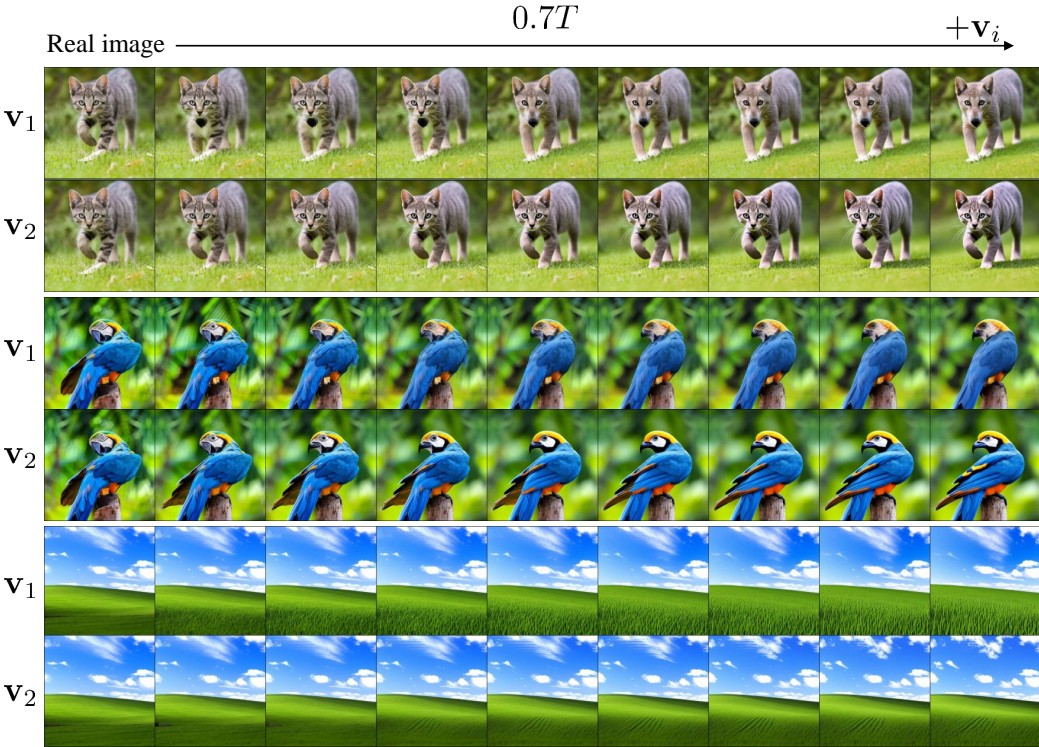

Figure A9: **More examples of image editing using the latent basis with Stable Diffusion.** The editing result using $\mathbf{v}_i$' in Stable diffusion at $0.7T$.

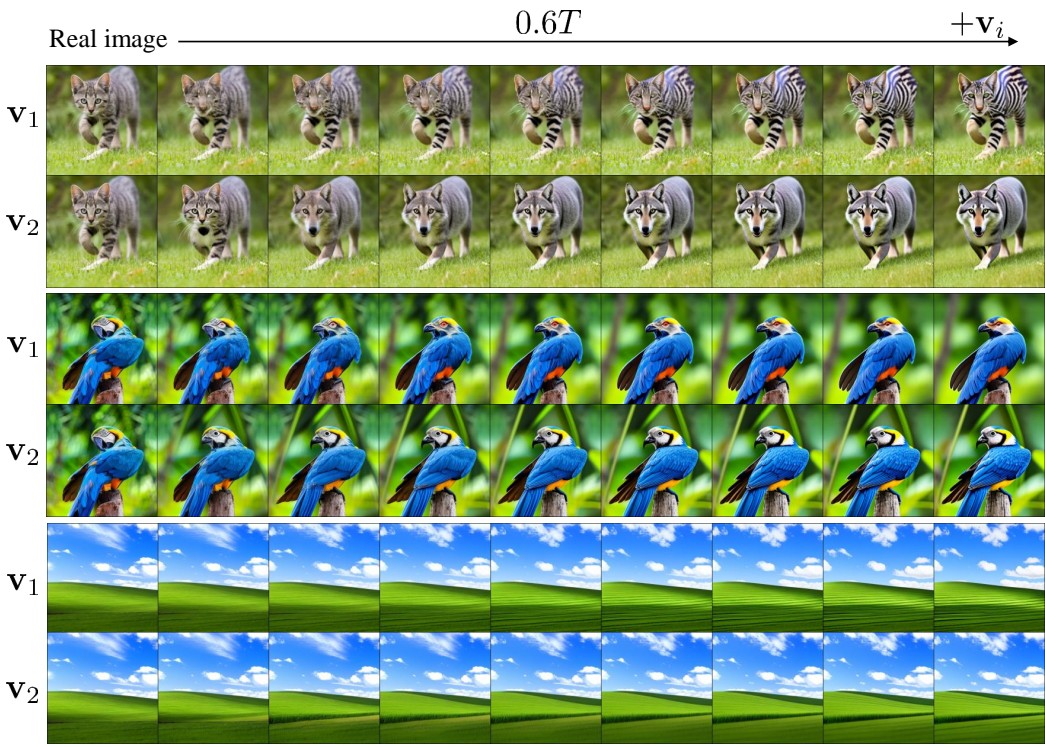

Figure A10: **More examples of image editing using the latent basis with Stable Diffusion.** The editing result using $\mathbf{v}_i$' in Stable diffusion at $0.6T$.

**latent basis with given prompt**   As shown in Figure A11, when we condition a specific prompt, such as "Zebra" or "Chimpanzee", the entire latent basis corresponds to the prompt-related attributes. Notably, Changes to "zebra", which are clear, all show similar results, but "chimpanzee" show different results. Nevertheless, it is clear that they are all related to "chimpanzee".

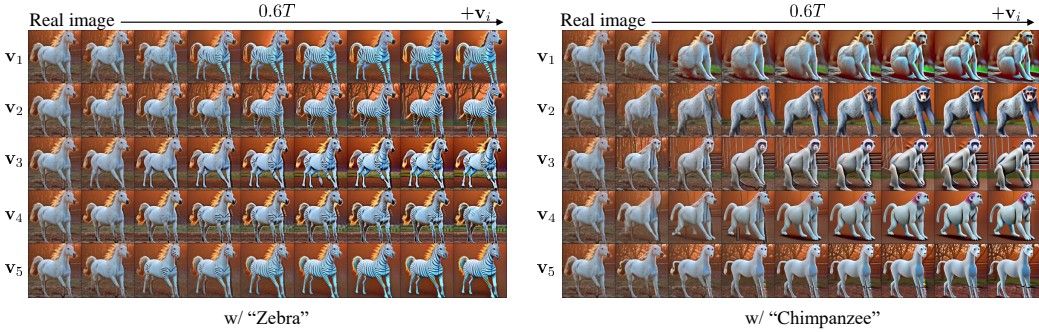

Figure A11: **More examples of image editing using top-5 latent basis vectors when given the prompt.** Notably, Changes to "zebras", which are clear, all show similar results, but "chimpanzees" show different results. Nevertheless, it is clear that it is all related to "chimpanzees".

## G.2    Image editing using latent basis vectors discovered with various prompts

We provide additional examples of image editing using latent basis vectors discovered with various prompts. Figure A12, A13 show image editing with various pictures and various prompts. For brevity, we denote the prompt "a cat dressed as a witch wearing a wizard hat in a haunted house" by "[ · · · cat · · ]" in Figure A12.

Real image ⟶ +$\mathbf{v}_i$

"Zebra"

"Ostrich"

"Crying dog"

"Koala"

"Chimpanzee"

"Horse"

[··· cat···] → [··· girl···]

[··· cat···] → [··· dog···]

"Eiffel Tower Fireworks"

"Eiffel Tower Full moon"

"Angelic"

"Demonic"

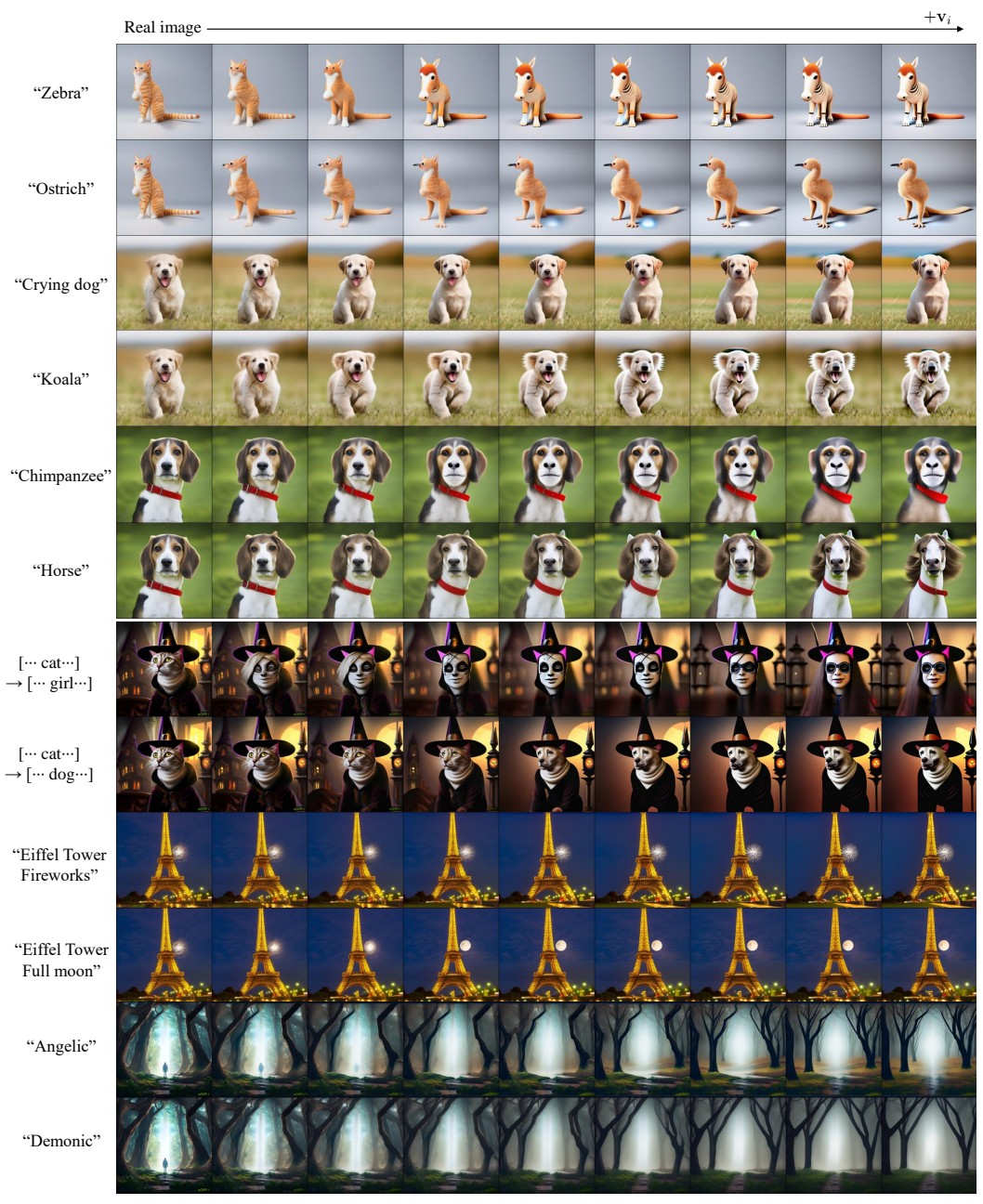

Figure A12: **More examples of image editing using latent basis vectors discovered with various prompts.**

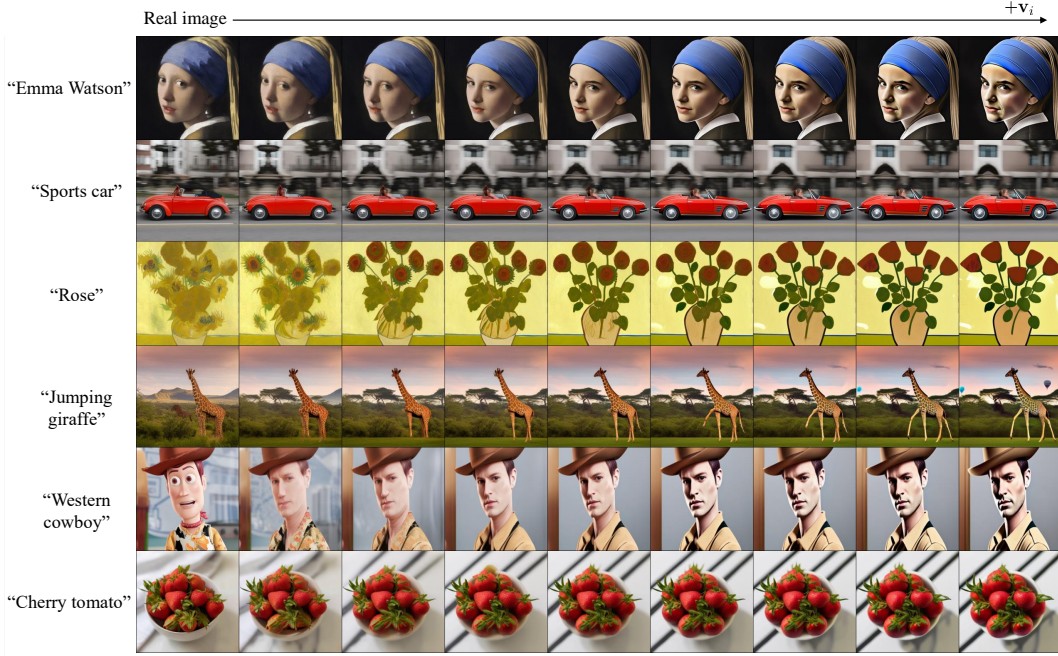

Figure A13: **More examples of image editing using latent basis vectors discovered with various prompts.**

### G.3 More discussion on the editing capability of the latent basis discovered with text conditions

In this subsection, we provide a discussion based on the failure cases of our approach. Figure A14 shows the results of latent basis found at $t = T$ with Stable diffusion. Unlike unconditional models, the directions found at $t = T$ exhibit rapid and drastic unexpected changes. However, landscape photos, which do not contain a main object, exhibited desired editing effects at any timesteps. Moreover, in the case of landscapes, it is conjectured that the latent basis plays a significant role in representing patterns and textures. Analyzing the landscape images generated by Stable diffusion would be an interesting topic.

Figure A15 presents examples of failure cases in our image editing using latent basis vectors discovered with various prompts. (a) When using pose or action as a prompt, there are instances where the identity is not preserved. (b) When the shape of the target subject differs significantly from the source image, the results are often unsatisfactory. (c) There are cases where the preservation of the background is not achieved. (d) It is challenging to make significant changes to the entire image.

Regarding the reasons for these failure cases, we emphasize two factors. First, we manipulate in the $\mathcal{X}$. The result in Figure A15 (a) implies that $\mathcal{X}$ is not a space where disentanglement for identity is achieved effectively. On the other hand, in $\mathcal{H}$, there are results indicating successful preservation of identity [26, 19]. Investigating the disentanglement capability of $\mathcal{X}$ and any other distinguishing features it may have compared to $\mathcal{H}$ would be an interesting future research topic.

Secondly, we perform manipulation by adding and subtracting the "signal" that the model pays attention to in $\mathbf{x}_t$. Here, The signal is captured from the current input $\mathbf{x}_t$, which limits the deviation from the original form. Therefore, when there is a substantial difference in shape, such as transforming a giraffe into a tiger, the results may not be satisfactory. (Figure A15 (b)) When we utilize text conditions, the latent basis aligns with the text information. This leads to not capturing background information, resulting in changes in the background when manipulated. It is also an interesting research topic to capture signals related to the background. (Figure A15 (c)) Since the model focuses on finer features as t approaches 0, if broad changes are desired, manipulation should be performed at

$t = T$. However, manipulation at $t = T$ is unstable. Deep analysis of $\mathbf{x}_t$ at $t = T$ in Stable diffusion is also an intriguing research topic. (Figure A15 (d))

Despite these limitations, we have successfully achieved direct manipulation in the latent space $\mathbf{x}_t$ at a single timestep, which, to our knowledge, is the first of its kind. Through this, we provide insights into the model and contribute to the understanding of the latent space, hopefully benefiting the diffusion community.

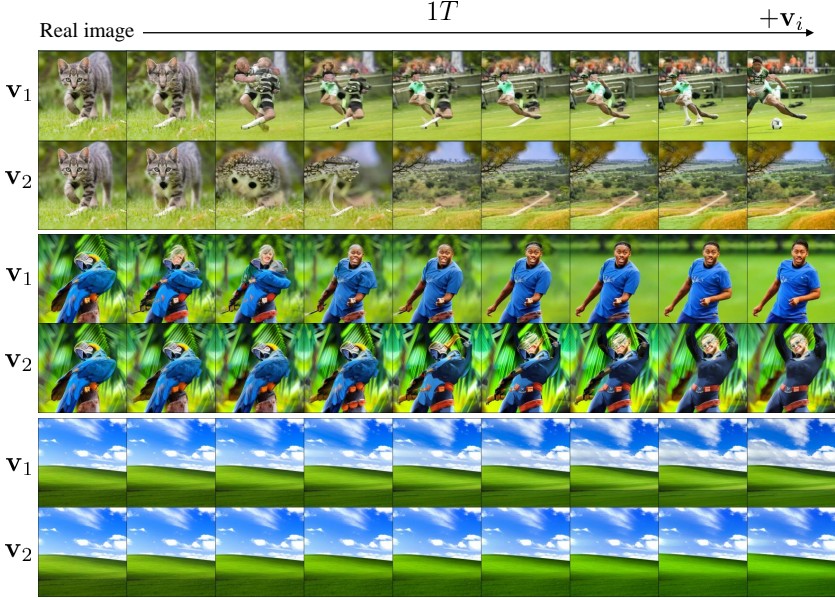

Figure A14: **Failure cases of image editing using the latent basis at $1T$.** The editing result using $\mathbf{v}_i$' in Stable diffusion at $1T$.

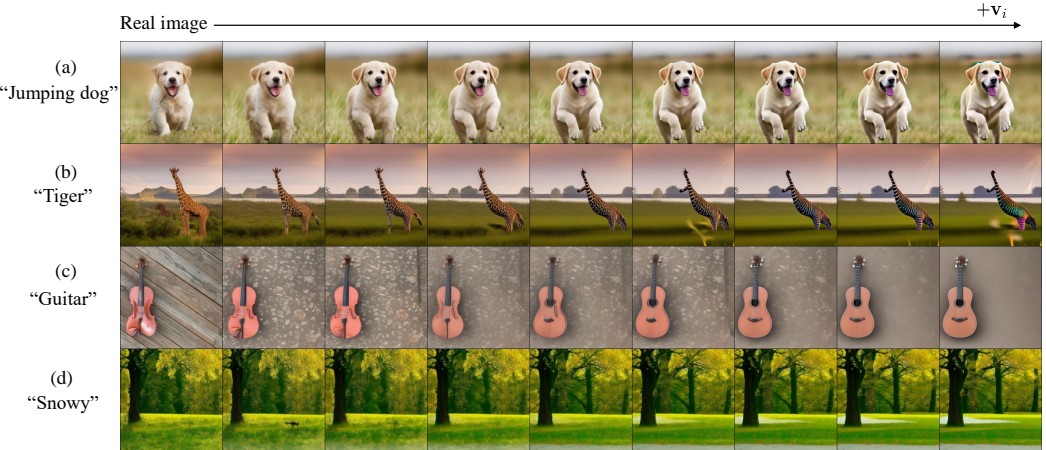

Figure A15: **Failure cases of using prompts.** (a) When using pose or action as a prompt, there are instances where the identity is not preserved. (b) When the shape of the target subject differs significantly from the source image, the results are often unsatisfactory. (c) There are cases where the preservation of the background is not achieved. (d) It is challenging to make significant changes to the entire image.