# OpenReview forum: "Understanding the Latent Space of Diffusion Models through the Lens of Riemannian Geometry"
_NeurIPS.cc/2023/Conference — NeurIPS 2023 poster_

### Official Review · Reviewer_iEgM · 2023-07-05

**Soundness:** 3 good
**Presentation:** 2 fair
**Contribution:** 3 good
**Rating:** 6
**Confidence:** 4

**Summary:**

The paper presents an analysis of the latent structure of diffusion models using differential geometry. The authors propose a method to define a geometry in the latent space by pulling back the Euclidean metric from the U-Net bottleneck space *H* via the network encoder. This approach enables the identification of directions of maximum variation in the latent space. The paper also explores the application of the proposed latent structure guidance for image editing. Finally, the evolution of the geometric structure over time steps and its dependence on text conditioning are investigated.

**Strengths:**

- I found the analysis presented in the paper interesting. It both uncovers unknown details about diffusion models (effect of text prompt and complexity of the dataset on the latent space) and confirm some previous observations (e.g., coarse-to-fine behaviour). This exploration can potentially reveal new capabilities of diffusion models, contributing to the advancement of the field.
- The paper is technically sound, and the claims made by the authors in sections 4 are supported by experiments. This experimental validation enhances the credibility of the proposed approach.

**Weaknesses:**

- The paper lacks comparisons with other diffusion-based image editing techniques, like [7,18]. Including such comparisons would have provided a more comprehensive evaluation and demonstrated the advantages of the proposed method.
- The presentation and clarity of the paper could be improved. For example, the abstract contains too much detail, making it challenging to understand upon initial reading. Also the explanation of the image editing technique could be improved: what is DDIM (section 4)? what is epsilon in Equation 4? Finally, Figure 1 is not sufficiently clear to me, it may hampers the reader's comprehension.

**Questions:**

To make the paper applicable to real-world scenarios right away, the authors should include a comparative analysis with other image editing techniques.

**Some potentially interesting references**
- Pan, Xingang, et al. "Drag Your GAN: Interactive Point-based Manipulation on the Generative Image Manifold." arXiv preprint arXiv:2305.10973 (2023).
- Brooks, Tim, Aleksander Holynski, and Alexei A. Efros. "Instructpix2pix: Learning to follow image editing instructions." Proceedings of the IEEE/CVF Conference on Computer Vision and Pattern Recognition. 2023.
- Meng, Chenlin, et al. "Sdedit: Guided image synthesis and editing with stochastic differential equations." arXiv preprint arXiv:2108.01073 (2021).

**Typos**
- line 20: double citation
- line 160 editted
- Figure 2 caption: editing

**Limitations:**

The authors have addressed the limitations of their approach.

---

> ### Author Rebuttal · Authors · 2023-08-09
>
> Thank you for acknowledging our strengths:
>
> - uncovering the effect of text prompt and dataset complexity on the latent space.
> - confirming coarse-to-fine behavior of diffusion models (DMs).
> - enhancing credibility through experimental validation.
>
> ---
>
> > [W1] The paper lacks comparisons with other diffusion-based image editing techniques.
>
> > [Q1] To make the paper applicable to real-world scenarios right away, the authors should include a comparative analysis with other image editing techniques.
>
> [W1, Q1] Thanks for the great suggestion. We conduct qualitative comparisons with text-guided image editing methods. Our state-of-the-art baseline methods include: (i) [SDEdit], (ii) [Pix2Pix-zero], (iii) [PnP], and (iv) [Instruct Pix2Pix]. All comparisons were performed using the official code. Please refer to the global rebuttal paragraph 3 and Fig. R3 in the global rebuttal PDF for the results.
>
> ---
>
> > [W2] The presentation and clarity of the paper could be improved. [W2-a] For example, the abstract contains too much detail, making it challenging to understand upon initial reading. [W2-b] Also the explanation of the image editing technique could be improved: [W2-b-i] What is DDIM (section 4)? [W2-b-ii] What is epsilon in Equation 4? [W2-b-iii] Finally, Figure 1 is not sufficiently clear to me, it may hamper the reader's comprehension.
>
> [W2-a] Thank you for your constructive suggestion. We revise the abstract to be concise and digestible as follows:
>
> >> Despite the success of diffusion models (DMs), we still lack a thorough understanding of their latent space. To understand the latent space $\mathbf{x}_t \in \mathcal{X}$, we analyze them from a geometrical perspective. Our approach involves deriving the local latent basis within $\mathcal{X}$ by leveraging the pullback metric associated with their encoding feature maps. Remarkably, our discovered local latent basis enables image editing capabilities by moving $\mathbf{x}_t$, the latent space of DMs, along the basis vector at specific timesteps.
> >>
> >> We further analyze how the geometric structure of DMs evolves over diffusion timesteps and differs across different text conditions. This confirms the known phenomenon of coarse-to-fine generation, as well as reveals novel insights such as the discrepancy between $\mathbf{x}_t$ across timesteps, the effect of dataset complexity, and the time-varying influence of text prompts. To the best of our knowledge, this paper is the first to present image editing through $\mathbf{x}$-space traversal, editing only once at specific timestep $t$ without any additional training, and providing thorough analyses of the latent structure of DMs.
>
>
> [W2-b] Thank you for pointing out the clarity issue.
>
> [W2-b-i] To make the image editing process easier to understand, we created a summary subsection and an overview figure for the whole process. Please refer to global rebuttal paragraph 2 and Fig. R2 in the global rebuttal PDF for more details. In our method, [DDIM] is used to invert the image into the initial noise $\mathbf{x}_T$, and again for denoising to generate the image.
>
>
> [W2-b-ii] $\epsilon$ is a function representing the pretrained diffusion model. For clarity, in the revised manuscript we will write it as $\epsilon^{\theta}$, and state it explicitly as below :
>
> >> (L166) ... where $\epsilon^{\theta}$ is the denoising function of the pretrained diffusion model, and $\gamma$ is ...
>
> [W2-b-iii] Thank you for the valuable suggestion. Please refer to global rebuttal paragraph 1 and Fig. R1, R2 in the PDF for improvements to Fig. 1. Below is an excerpt from the content of the global rebuttal paragraph 1, extracted to enhance readability for the reviewers:
>
> In order to make Fig. 1 more comprehensible, we divide Fig. 1 into two separate figures. Please refer to Fig. R1 and R2 in the global rebuttal PDF. First, **Figure R1** conceptually visualizes the local basis found through the pullback metric and provides a summary of the process for obtaining a local basis. Second, **Figure R2** provides an overview of the image editing method using the discovered local basis and briefly presents its results.
>
> ---
>
> > Some potentially interesting references
>
> Thank you for your valuable references. We will add those suggested references to the revised manuscript.
>
> > Typos
>
> Thank you for the careful comments. We will address and incorporate these revisions in the upcoming manuscript.
>
>
> ---
>
> **References**
>
> [SDEdit] : SDEdit: Guided Image Synthesis and Editing with Stochastic Differential Equations, Meng et al., 2021
>
> [Pix2Pix-zero] : Zero-shot Image-to-Image Translation, Parmar et al., 2023
>
> [PnP] : Plug-and-Play Diffusion Features for Text-Driven Image-to-Image Translation, Tumanyan et al., 2022
>
> [Instruct Pix2Pix] : InstructPix2Pix: Learning to Follow Image Editing Instructions, Brooks et al., 2022
>
> [DDIM] : Denoising Diffusion Implicit Models, Song et al., 2020

---

> > ### Author Response · Authors · 2023-08-19
> >
> > We sincerely thank you for your first review once again.
> >
> > As the discussion phase ends soon, we remain enthusiastic about receiving additional feedback from you. We are ready to accommodate your needs if you find our revised response requires additional clarifications and suggestions.
> >
> > Thank you.

---

> > > ### Comment · Area_Chair_CbYX · 2023-08-21
> > > **Thank you for your rebuttal**
> > >
> > > Dear authors,
> > > Thank you for your rebuttal and clarifications -- this supports us as we assess the paper and its reviews during the discussion and decision phase.
> > >
> > > Thanks,
> > > Your AC

---

> > > ### Comment · Reviewer_iEgM · 2023-08-21
> > >
> > > I thank the reviewer for their time and effort in making the rebuttal and answering my questions. Overall, I am satisfied with the additional comparison and additional information about time complexity.
> > > After having read the other reviews and the author's response, I lean towards acceptance.

---

> > > > ### Author Response · Authors · 2023-08-22
> > > > **Thank you!**
> > > >
> > > > Again, thank you for your feedback and your great efforts. Any further questions/suggestions would be also appreciated.

---

### Official Review · Reviewer_6jtZ · 2023-07-05

**Soundness:** 2 fair
**Presentation:** 1 poor
**Contribution:** 3 good
**Rating:** 4
**Confidence:** 4

**Summary:**

In this submission, the authors probe the latent space, xt ∈ X, of diffusion models (DMs) from a geometric perspective, utilizing the pullback metric to identify local latent basis in X and corresponding local tangent basis in H. To confirm their findings, they edit images via latent space traversal. The authors provide a two-pronged analysis, investigating the evolution of geometric structure over time and its variation based on text conditioning in Stable Diffusion. Notably, they discovered that in the generative process, the model prioritizes low-frequency components initially, moving to high-frequency details later, and that the model's dependence on text conditions reduces over time. The paper introduces image editing through x-space traversal and to offer comprehensive analyses of the latent structure of DMs, with a specific emphasis on the use of the pullback metric and the SVD of the Jacobian in computing a basis.

**Strengths:**


The paper presents a distinctive idea that provides an alternative method for editing in diffusion models, as well as enhancing comprehension of the latent space dynamics. By utilizing a geometric perspective, the authors make use of the pullback metric to investigate the latent space, offering insights into its structure and operation.  The exploration of the evolution of geometric structure over time and its response to various text conditions offers additional insights into the dynamics of the latent space is interesting.

**Weaknesses:**


The first area where the paper could see improvement is in terms of the clarity of its analysis. Given its nature as an analysis paper, it's crucial that the analysis presented is as comprehensible as possible. However, the method and notation used in this work can lead to some confusion. For instance, Section 3, in its current form, may not be as accessible to all readers as it could be, and it could benefit from being revised for clearer communication of the ideas contained therein. Additionally, Fig. 1, which is presumably intended to illustrate key concepts, is perhaps too dense with information. Dividing Fig. 1 into two separate figures could make it easier to digest, enabling a clearer explanation of the approach.

A second aspect that could be improved upon is the overall presentation and proofreading of the paper. While the approach is relatively simple, its translation into the written form has not been as clear as one would hope. The text could benefit from a thorough proofreading to ensure that it is not just grammatically correct, but also that it conveys the authors' ideas in a way that is accessible to the wider machine learning community. As it stands, the paper's usefulness to this community may be hindered by its presentation.

Lastly, the paper could do more to address the computational implications of its approach. The authors use the power method to approximate the Jacobian, which, while effective, can be computationally costly. It would be beneficial if the authors were more transparent about this fact, allowing readers to fully understand the computational demands of the approach and evaluate whether or not it would be feasible in their own applications. Being upfront about such limitations can help to build a more honest and comprehensive understanding of the paper's methodologies and implications.

**Questions:**

What is the complexity of computing the Jacobian? What is the the actual runtime (in seconds) of the approach compared to other editing methods? I think answering these questions can provide a good context for readers.

**Limitations:**

Societal impact has been properly address but I would like to see a deeper analysis of the computational complexity and runtime of the approach.

---

> ### Author Rebuttal · Authors · 2023-08-09
>
> Thank you for acknowledging the strengths of our paper: the distinctive idea for editing in diffusion models (DMs), and enhancing comprehension of the latent space dynamics, e.g., the evolution of geometric structure over time and the influence of various text conditions.
>
> ---
>
> > [W1] ... the method and notation used in this work can lead to some confusion. For instance, [W1-a] Section 3, in its current form, may not be as accessible to all readers as it could be, and it could benefit from being revised for clearer communication of the ideas contained therein. [W1-b] Additionally, Fig. 1, which is presumably intended to illustrate key concepts, is perhaps too dense with information. Dividing Fig. 1 into two separate figures could make it easier to digest, enabling a clearer explanation of the approach.
>
> [W1] Thank you for the constructive suggestion.
>
> [W1-a] To clarify the image editing process in Section 3, we added a new subsection summarizing the overall procedure. This includes a visual overview in Fig. R2 and clear textual explanations. Please refer to global rebuttal 2 for a more detailed description.
>
> [W1-b] To make Fig. 1 more comprehensible, we divide it into two figures, Fig. R1 and R2. Please refer to the global rebuttal 1 for more detailed improvements.
>
> ---
>
> > [W2] A second aspect that could be improved upon is the overall presentation and proofreading of the paper. While the approach is relatively simple, its translation into the written form has not been as clear as one would hope. The text could benefit from a thorough proofreading to ensure that it is not just grammatically correct, but also that it conveys the authors' ideas in a way that is accessible to the wider machine learning community. As it stands, the paper's usefulness to this community may be hindered by its presentation.
>
> [W2] Thank you for the advice on readability. We made the following improvements to help readers understand more easily:
>
> + We relocate the discussion on parallel transport (P.T.) from section 3.3 to the final subsection of section 3. This alteration was implemented to prevent any potential misunderstanding that using P.T. is the default approach in our editing method. By introducing P.T. as a special case after concluding the explanation of the image editing method, we aim to mitigate such misconceptions.
>
> + We consolidate all elements related to the image editing method within section 3. For instance, the process of generating initial noise $\mathbf{x}_T$ through DDIM inversion, previously outlined in section 4 (L178-182), has been now moved to the newly introduced subsection referenced in global rebuttal 2. This organization ensures that readers can fully comprehend the entire editing process by solely referring to section 3.
>
> + We make adjustments to Figure 8. Specifically, we have relocated Fig. 8(b) and the content spanning from L243 to L248 to the appendix. This decision was based on the rationale that Figure 8(b) primarily serves to validate the findings in (a) and (c). Including this information in the appendix was deemed a more suitable arrangement.
>
> We will also carefully identify further shortcomings and revise them in the camera-ready version.
>
> ---
>
> > [W3] Lastly, the paper could do more to address the computational implications of its approach. The authors use the power method to approximate the Jacobian, which, while effective, can be computationally costly. It would be beneficial if the authors were more transparent about this fact, allowing readers to fully understand the computational demands of the approach and evaluate whether or not it would be feasible in their own applications. ...
>
> > [L1] I would like to see a deeper analysis of the computational complexity and runtime of the approach.
>
> [W3, L1] In Appendix A, we present the computation time required for approximating the Singular Value Decomposition (SVD) of the Jacobian using the power method. The runtime of this power method varies based on the parameter $n$, which denotes the low-rank approximation of the original tangent space. Smaller values of $n$ lead to shorter computation times. For instance, when $n=3$, the process takes approximately 10 seconds. In contrast, for $n=50$, the computation time extends to around 3 to 4 minutes, particularly in the context of Stable Diffusion.
>
> Notably, when only a single basis vector is needed, as is the case in scenarios like text conditional editing, the time taken to approximate the SVD of the Jacobian is remarkably brief—approximately 2.5 seconds.
>
> ---
>
> > [Q1] What is the complexity of computing the Jacobian? What is the actual runtime (in seconds) of the approach compared to other editing methods? I think answering these questions can provide a good context for readers.
>
> [Q1] Thank you for your valuable question. We conduct all comparisons on the Nvidia 3090. To ensure a fair comparison, we set $n=1$ and performed 50 steps of the DDIM algorithm. The time taken by each method is as follows:
>
> | Image Edit Method | running time  | Preprocessing |
> |:-----------------:|:-------------:|:-------------:|
> |  Ours             |     11 sec    |N/A            |
> |  SDEdit           |      4 sec    |N/A            |
> |  Pix2Pix-zero     |     25 sec    |4 min*             |
> |  PnP              |     10 sec    |40 sec**  |
> |  Instruct Pix2Pix |     11 sec    |N/A     |
>
> (*: generating 100 prompts with GPT, obtaining embedding vectors, **: storing feature vectors, queries, and key values)
>
> For a more detailed discussion, please refer to global rebuttal paragraph 3.
>
> ---
>
> **References**
>
> [SDEdit] : SDEdit: Guided Image Synthesis and Editing with Stochastic Differential Equations, Meng et al., 2021
>
> [Pix2Pix-zero] : Zero-shot Image-to-Image Translation, Parmar et al., 2023
>
> [PnP] : Plug-and-Play Diffusion Features for Text-Driven Image-to-Image Translation, Tumanyan et al., 2022
>
> [Instruct Pix2Pix] : InstructPix2Pix: Learning to Follow Image Editing Instructions, Brooks et al., 2022

---

> > ### Comment · Reviewer_6jtZ · 2023-08-15
> >
> > I have read the rebuttal and thank the authors for their answers. Given the new details provided by authors in terms of computational complexity and runtime of their approach I cannot update my score.

---

> > > ### Author Response · Authors · 2023-08-15
> > >
> > > Thank you for dedicating time to share your thoughts.
> > >
> > > Firstly, we would like to mention that the main contribution of our paper is the geometric interpretation of the latent space and feature space. To the best of our knowledge, the intricate interplay between semantics and the geometric structure within the latent space of diffusion models remains unexplored. We believe that our paper will significantly enrich future research. \
> > > (Also, please consider the other points written in the global comment.)
> > >
> > > Moreover, we highlight that our method achieves a comparable complexity, even though efficiency was not our primary focus. \
> > > (We would like to kindly clarify that the 100 seconds mentioned in Reviewer vuQy's rebuttal were used for computing all 50 local bases for analysis. Additionally, we wish to mention again that the reported 11 seconds include the computation time for the 1st local basis. Because SDEdit is the oldest basic, do-nothing approach, our method has comparable computational complexity except for SDEdit.)
> > >
> > > Would you reconsider these points?

---

### Official Review · Reviewer_vuQy · 2023-07-05

**Soundness:** 2 fair
**Presentation:** 2 fair
**Contribution:** 2 fair
**Rating:** 5
**Confidence:** 5

**Summary:**

This paper studies the geometry of latent spaces of diffusion models (Dms) using the pullback metric. In the analyses, they mainly examine change of the frequency representations in latent spaces over time and the change of the structure based on text conditioning.

After the rebuttal:

I checked all reviewer comments and responses.

I agree with the other reviewers regarding limited algorithmic novelty of the work. Therefore, I keep my original score.

**Strengths:**

- The paper is well written in general (there are several typos but they can be fixed in proof-reading).

- The proposed methods and analyses are interesting. Some of the theoretical results highlight several important properties of diffusion models.

**Weaknesses:**

1. Some of the statements and claims are not clear as pointed in the Questions.

2. The results are given considering the Riemannian geometry of the latent spaces and utilizing the related transformations (e.g. PT) among tangent spaces on the manifolds. However, vanilla DMs do not employ these transformations. Therefore, it is not clear whether these results are for vanilla DMs or the  DMs utilizing the proposed transformations.

3. A major claim is that the proposed methods improve effectiveness of the DMs. However, the employed transformations can increase the footprints of DMs.

**Questions:**

It is stated that “To investigate the geometry of the tangent basis, we employ a metric on the Grassmannian manifold.” However, the space could be identified by another manifold as well.  Why and how did you define the space by the Grassmannian manifold?

It is claimed that “the similarity across tangent spaces allows us to effectively transfer the latent basis from one sample to another through parallel transport”. How this improves effectiveness was not analyzed. In general, how do the proposed methods improve training and inference time? Indeed, the additional transformations can increase training and inference time. Could you please provide an analysis of the footprints?

**Limitations:**

Some of the limitations were addressed but potential impacts were not addressed.

---

> ### Author Rebuttal · Authors · 2023-08-09
>
> > [W1] Some of the statements and claims are not clear as pointed in the Questions.
>
> > [Q1] It is stated that "we employ a metric on the Grassmannian manifold.” Why and how did you define the space by the Grassmannian manifold?
>
> [Q1] These subspaces (as a vector space) of $\mathcal{T_{\mathbf{x}}}$ and $\mathcal{T_{\mathbf{h}}}$ assigned at the points $\mathbf{x} \in \mathcal{X}$ and $\mathbf{h} \in \mathcal{H}$ of diffusion models. Given that the Grassmannian manifold is characterized as a manifold comprising subspaces, it appears well-suited for representing the manifold of $\mathcal{T_{\mathbf{h}}}$. Additionally, the geodesic metric defined on this manifold quantifies the separation between two subspaces based on their principal angles. Hence, we posit that this geodesic metric offers a robust means of evaluating the similarity between different $\mathcal{T_{\mathbf{h}}}$s.
>
> > [Q2-a] It is claimed that “the similarity across tangent spaces allows us to effectively transfer the latent basis from one sample to another through parallel transport”. How this improves effectiveness was not analyzed. [Q2-b] In general, how do the proposed methods improve training and inference time? (...) Could you please provide an analysis of the footprints?
>
> [Q2-a] Figure 7 shows the impact of employing parallel transport (P.T.). The transported vectors can edit distinct samples while manipulating the same attributes. We will modify L227-299 as follows to better reflect our intent:
>
> >> (L227-229) $\rightarrow$ ... us to successfully transfer the latent basis vector from one sample to another through parallel transport. Figure 7 shows that the transported vectors can edit distinct samples while manipulating the same attributes.
>
> [Q2-b] We would like to emphasize that our approach does not require any form of training. Furthermore, the utilization of P.T. is not a default procedure in the editing process; its application is limited to specific instances where local basis vectors from other samples are used. The details regarding the typical editing footprint in the absence of P.T. are outlined in the global rebuttal 3. Here, for your reference, we focus solely on presenting the result table:
>
> |Image Edit Method|running time|Preprocessing|
> |:-:|:-:|:-:|
> |Ours| 11 sec |N/A|
> |[SDEdit]|4 sec|N/A|
> |[Pix2Pix-zero]|25 sec|4 min|
> |[PnP]|10 sec|40 sec|
> |[Instruct Pix2Pix]|11 sec|N/A|
>
> The time taken for editing using P.T. can be broken down into three components: 1) DDIM inversion and generation, 2) identification of the local basis, and 3) the parallel transport process. For optimal results when employing P.T., it is advisable to use a sufficiently large value of $n$ to mitigate distortion. However, this choice necessitates more computation time for generating the local basis. In our case, we opt for $n=50$, with the number of DDIM steps set at $100$, and utilizing an unconditional diffusion model trained on CelebA-HQ.
>
> |DDIM inversion + generation|Identification of local basis|Parallel transport|
> |:-:|:-:|:-:|
> | 10sec | 100sec | 0.002sec|
> ---
> > [W2] The results are given ... utilizing the related transformations (e.g. PT) among tangent spaces on the manifolds. However, vanilla DMs do not employ these transformations. Therefore, it is not clear whether these results are for vanilla DMs or the DMs utilizing the proposed transformations.
>
> [W2] We wish to emphasize that our method works on frozen vanilla DMs without necessitating any fine-tuning or architectural modifications. It offers unsupervised image editing capabilities that are applicable to both unconditional DMs and conditional DMs. Furthermore, as explained in [Q2], it is important to note that the inclusion of P.T. is not a default step in our editing process. This technique is selectively employed in particular cases where local basis vectors are transferred for editing other samples.
>
> > [W3] A major claim is that the proposed methods improve the effectiveness of the DMs. However, the employed transformations can increase the footprints of DMs.
>
> [W3] To provide clarification, our primary focus centers on image editing utilizing DMs, without an emphasis on augmenting the performance of DMs themselves.
>
> When performing image editing with a single latent basis vector (i.e., $n=1$), our editing process consumes approximately 11 seconds. This represents roughly 15% of the time required for vanilla inversion and reconstruction. Furthermore, even when employing Jacobian approximation with $n=1$, the computational overhead associated with our approach remains comparable with other state-of-the-art editing methods, as illustrated in the aforementioned table.
>
> > [L1] Some of the limitations were addressed but potential impacts were not addressed.
>
> [L1] We appreciate your observation and would like to address this concern by incorporating a societal impact and ethics statement into the revised manuscript, which is presented below:
>
> >> **Societal Impact / Ethics Statement.** Our research endeavors to unravel the geometric structures of the diffusion model and facilitate high-quality image editing within its framework. While our primary application resides within the creative realm, it is important to acknowledge that image manipulation techniques, such as the one proposed in our method, hold the potential for misuse, including the dissemination of misinformation or potential privacy implications. Therefore, the continuous advancement of technologies aimed at thwarting or identifying manipulations rooted in generative models remains of utmost significance.
>
> **References**
>
> [SDEdit] : SDEdit: Guided Image Synthesis and Editing with Stochastic Differential Equations, Meng et al., 2021
>
> [Pix2Pix-zero] : Zero-shot Image-to-Image Translation, Parmar et al., 2023
>
> [PnP] : Plug-and-Play Diffusion Features for Text-Driven Image-to-Image Translation, Tumanyan et al., 2022
>
> [Instruct Pix2Pix] : InstructPix2Pix: Learning to Follow Image Editing Instructions, Brooks et al., 2022

---

> > ### Author Response · Authors · 2023-08-19
> >
> > We sincerely thank you for your first review once again.
> >
> > As the discussion phase ends soon, we remain enthusiastic about receiving additional feedback from you. We are ready to accommodate your needs if you find our revised response requires additional clarifications and suggestions.
> >
> > Thank you.

---

> > > ### Comment · Area_Chair_CbYX · 2023-08-21
> > > **Thank you for your rebuttal**
> > >
> > > Dear authors,
> > > Thank you for your rebuttal and clarifications -- this supports us as we assess the paper and its reviews during the discussion and decision phase.
> > > Thanks,
> > > Your AC

---

> > > ### Comment · Reviewer_vuQy · 2023-08-21
> > >
> > > Thank you for the response to the questions.
> > >
> > > I checked all reviews and responses. Since my questions are addressed, I upgrade my score. However, I agree with the other reviewers that the novelty of the paper should be improved considering its limitations in theory and practical application, i.e. footprints, for a strong acceptance.

---

> > > > ### Author Response · Authors · 2023-08-22
> > > > **Thank you!**
> > > >
> > > > Again, thank you for your feedback and your great efforts. Any further questions/suggestions would be also appreciated.

---

> > > > ### Author Response · Authors · 2023-08-22
> > > >
> > > > Thank you for dedicating time to share your thoughts.
> > > >
> > > > We would like to highlight that our method achieves a comparable complexity, even though efficiency was not our primary focus.
> > > > (We would like to kindly clarify that the 100 seconds mentioned in the response were used for computing all 50 local bases for parallel transport, not for the default editing process. Additionally, we wish to mention again that the reported 11 seconds include the computation time for the 1st local basis. Because SDEdit is the oldest basic, do-nothing approach, our method has comparable computational complexity except for SDEdit.)
> > > >
> > > > Would you reconsider these points?

---

### Official Review · Reviewer_cKEc · 2023-07-10

**Soundness:** 3 good
**Presentation:** 2 fair
**Contribution:** 3 good
**Rating:** 7
**Confidence:** 3

**Summary:**

The paper proposes a study on the latent space of diffusion models and on how to manipulate it. It takes advantage of an observation made by previous work [22] on the flatness and semantic structure of the U-Net model used in  DDIM and uses pullback metric from the latent space of the U-Net to the space of diffusion to measure some properties of the latter under different conditions.
The paper also proposes a method to manipulate the diffusion space through the different time steps, so as to carry out semantically meaningful edits.

**Strengths:**

This paper is one of the first to study the behavior of the space diffusion models. It presents some interesting studies on the behavior of the process during time, showing that the early stages convey higher frequency while the last steps are more concerned with higher frequencies. Another interesting observation is that tangent spaces of different samples tend to be more aligned at T=1, while they diverge toward T=0 (end of the generative process).

The paper also shows (even if it had already been observed in 22) that the pullback metric is effective in transferring the shift along the semantically meaningful principal components of the U-Net latent space into the diffusion process, thus resulting in meaningful edits of the generated image, which frequency depends on the time the edit was performed.

**Weaknesses:**

I may have misunderstood or missed some important information, but the method described is not really clear. Specifically, it is not really clear to me how the editing process works (sections 3.3 and 3.4):
- In 3.3, the letter v is used to indicate elements of both T_x and T_h, so it is not always clear to which space they are referring.
- In general, it is not clear why the idea expressed in 3.3 is useful and where it was adopted.
- In eq 4 what is the epsilon function? In general, isn’t the vector toward which to shift selected from T_H (so it should be u) and then transferred to T_x?


Another concern is about the generalization of the proposed method to other diffusion techniques, or with other score models (i.e. not UNet). I think that this point needs more discussion.

**Questions:**

I was wondering what would happen if, instead of moving along one of the principal axis of T_H, you use directly the principal axis of T_x.

I would also discuss a bit more method 22 in the related work since it seems related to the proposed method.

Writing issues:
 - Check the sentence at rows 74-75
- Row 152: we aims
- general grammar check

**Limitations:**

I don’t foresee any particular negative societal impact. A discussion on how the proposed study may generalize to other domains and architectures would be of value.

---

> ### Author Rebuttal · Authors · 2023-08-09
>
> Thank you for acknowledging the strengths of our paper: first to study the behavior of the space of diffusion models (DMs), e.g., the coarse-to-fine behavior, the divergence of tangent space across different samples, and the meaningful image editing with pullback metric.
>
> ---
>
> > [W1] It is not really clear to me how the editing process works (sections 3.3 and 3.4)
>
> [W1] Please refer to the global rebuttal 2.
>
> > + [W1-a] In 3.3, the letter v is used to indicate elements of both T_x and T_h, so it is not always clear to which space they are referring.
>
> [W1-a] We thank you for identifying the typo, $\mathbf{v} \in \mathcal{T_\mathbf{h}}$, in Section 3.3 Line 148. It should be corrected to $\mathcal{T_\mathbf{h}} \rightarrow \mathcal{T_\mathbf{x}}$. The vector notations, $\mathbf{v}\in\mathcal{T_\mathbf{x}}$ and $\mathbf{u}\in\mathcal{T_\mathbf{h}}$ are specifically employed for each space.
>
> > + [W1-b] In general, it is not clear why the idea expressed in 3.3 is useful and where it was adopted.
>
> [W1-b] Parallel transport (P.T.) is a concept in differential geometry that involves transporting a vector between spaces while maintaining its direction relative to the space. We use the P.T. for two purposes:
>
> * First, we use it for image editing. Considering the inherent characteristics of the unsupervised image editing method, it becomes imperative to manually inspect the semantic relevance of the latent basis vector within the edited results. Let us consider a scenario where our aim is to edit ten images of straight hair into curly hair. If we do not use P.T., we have to manually find a straight-to-curly basis vector for individual samples. P.T. allows transporting a straight-to-curly basis vector in one sample to all other samples to edit all images with only one manual inspection.
>
> * Second, we employ it to verify the similarities among local geometrical structures across various samples as shown in Fig. 7. This empirical demonstration substantiates that the geodesic distance among local geometrical structures holds tangible significance rather than being merely a conceptual measure as depicted in Fig. 6.
>
> > + [W1-c-i] In eq 4 what is the epsilon function? [W1-c-ii] In general, isn’t the vector toward which to shift selected from T_H (so it should be u) and then transferred to T_x?
>
> [W1-c-i] $\epsilon$ is the denoising function of the pretrained DM. For clarity, we will explicitly state this as $\epsilon^{\theta}$ in the revised manuscript:
>
> >> (L166) ... where $\epsilon^{\theta}$ is the denoising function of the pretrained DM, and $\gamma$ is ...
>
> [W1-c-ii] Through the decomposition of the Jacobian, we can simultaneously obtain both $\mathbf{v}$ and its corresponding $\mathbf{u}$. However, we use only $\mathbf{v}$ during the editing procedure.
>
> ---
>
> > [W2] Another concern is about the generalization of the proposed method to other diffusion techniques, or with other score models (i.e. not UNet). I think that this point needs more discussion.
>
> > [L1] A discussion on how the proposed study may generalize to other domains and architectures would be of value.
>
> [W2, L1] Thank you for raising this important issue. We add the discussion in the revised manuscript as follows:
>
> >> Our approach exhibits broad applicability in cases where a feature space within the DM possesses a Euclidean metric, as exemplified by $\mathcal{H}$. This characteristic has been observed in the context of U-Net within [Asyrp]. The question of whether alternative architectures, such as those resembling the structures of [DiT] or [MotionDiffusion], could also manifest a Euclidean metric, presents an intriguing avenue for future investigation.
>
> ---
>
> > [Q1] I was wondering what would happen if, instead of moving along one of the principal axis of T_H, you use directly the principal axis of T_x.
>
> [Q1] As previously mentioned in [W1-c-ii], we directly manipulate the latent variable $\mathbf{x}_t$ with the discovered latent basis vector $\mathbf{v}$. This approach offers several advantages over the editing of $\mathcal{H}$ as described in [Asyrp]. First, direct control over $\mathbf{h}_t$ disrupts the coherence between inner features and skip connections within the U-Net, which can lead to artifacts, particularly when making substantial adjustments [DiffStyle]. As a result, [Asyrp] implements gradual changes across multiple timesteps. In contrast, the manipulation of $\mathbf{x}_t$ enables more substantial alternations within a single timestep. Second, unlike $\mathbf{u}$ which exclusively affects the deepest feature map, $\mathbf{v}$ exerts influence not only on the latent variable $\mathbf{x}_t$ but also on all the feature map inside the DMs.
>
> ---
>
> > [Q2] I would also discuss a bit more method [25] in the related work since it seems related to the proposed method.
>
> [Q2] To enhance the distinction between our paper and [Asyrp], we modified the related work as follows:
>
> >> ... Kwon et al. [25] demonstrated that the bottleneck of the U-Net, $\mathcal{H}$, can be used as a semantic latent space. Specifically, they used CLIP to identify directions within $\mathcal{H}$ that facilitate genuine image editing. ... In contrast to the method, our editing approach involves the direct manipulation of latent variables within the latent space. Furthermore, we autonomously identify editing directions in an unsupervised manner.
>
> ---
>
> > [Q3] Writing issues:
> > + Check the sentence at rows 74-75
> > + Row 152: we aims
> > + general grammar check
>
> [Q3] Thank you for the careful comments. We will address and incorporate these revisions in the upcoming manuscript.
>
> **References**
>
> [Asyrp] : Diffusion Models already have a Semantic Latent Space, Kwon et al., 2022
>
> [DiT] : Scalable Diffusion Models with Transformers, Peebles et al., 2022
>
> [MotionDiffusion] : Human Motion Diffusion Model, Tevet et al., 2022
>
> [DiffStyle] : Training-free Style Transfer Emerges from h-space in Diffusion models, Jeong et al., 2023

---

> > ### Comment · Reviewer_cKEc · 2023-08-16
> > **discussion**
> >
> > Dear Authors,
> > thanks for taking the time to answer my doubts. I'm satisfied with the rebuttal and with the changes you promised to make in the revised manuscript.

---

> > > ### Author Response · Authors · 2023-08-19
> > > **Thank you!**
> > >
> > > Thank you for your feedback and your great efforts. Any further questions/suggestions would be also appreciated.
> > >
> > > Thank you.

---

### Author Rebuttal · Authors · 2023-08-09

We thank the reviewers for their valuable advice. Here, we compile reviews we want to share with all the reviewers. Please see our responses addressing the specific concerns below:

---

### 1. Improving Fig. 1 for clarity
**Reviewers *6jtZ* and *iEgM* suggested modifying Fig. 1 since it has too dense information.**

To enhance the comprehensibility of Fig. 1, we divide Fig. 1 into two figures. Please refer to Fig. R1 and R2 in the attached PDF file. **Figure R1** gives a conceptual visualization of the local basis derived from the pullback metric. This figure succinctly outlines the methods involved in obtaining the local basis. **Figure R2** delivers an overview of the image editing method using the discovered local basis. It also provides a concise presentation of the outcomes.

---

### 2. Additional subsection to overview the whole image editing process
**Reviewers *cKEc*, *6jtZ*, and *iEgM* asked the clear explanation of the editing process.**

We create a new subsection in section 3 that summarizes the entire procedure of image editing. This subsection provides clear explanations and also visually illustrates the method in Fig. R2. The detailed contents are as follows:

>> ### 3.4. The overall process of image editing
>> In this section, we summarize the entire editing process with five steps: 1) The input image is inverted into initial noise $\mathbf{x}_T$ using DDIM inversion. 2) $\mathbf{x}_T$ is gradually denoised until $t$ through DDIM generation. 3) Identify local latent basis $\{ \mathbf{v}_1, \cdots, \mathbf{v}_n \}$ using the pullback metric at $t$. 4) Manipulate $\mathbf{x}_t$ along the one of basis vectors using the $\mathbf{x}$-space guidance. 5) The DDIM generation is then completed with the modified latent variable $\tilde{\mathbf{x}}_t$. Figure R2 illustrates the entire editing process.
>>
>>In the context of a text-to-image model, such as Stable Diffusion, it becomes possible to include textual conditions while deriving local basis vectors. It aligns all the local basis vectors with the condition text. Comprehensive experiments can be found in Section 4.1.
>>
>>It is noteworthy that while our approach involves moving the latent variable within a single timestep, it achieves semantically meaningful image editing. In addition to the image manipulation within a single timestep, the direct manipulation of the latent variable, $\mathbf{x}_t$, of diffusion models is a pioneering approach to the best of our knowledge.


---


### 3. Comparative experiment to other state-of-the-art (SoTA) editing methods

**Reviewers *iEgM* and *6jtZ* provided constructive discussion about including comparative analysis for qualitative comparisons and runtime aspects.**

We conduct qualitative comparisons with text-guided image editing methods. Our state-of-the-art baseline methods include: (i) [SDEdit], (ii) [Pix2Pix-zero], (iii) [PnP], and (iv) [Instruct Pix2Pix]. All comparisons were performed using the official code. **Please refer to Fig. R3 in the PDF for the qualitative results.**


We also compare the time complexity of each method. For a fair comparison, we only identify the first singular vector $\mathbf{v}_1$, i.e., $n=1$, and set the number of DDIM steps to 50. All experiments were conducted on an Nvidia RTX 3090. The runtime for each method is as follows:

| Image Edit Method | running time  | Preprocessing |
|:-------------------------:|:------------------:|:-------------------:|
|  Ours             |     11 sec    |N/A           |
|  SDEdit           |      4 sec    |N/A            |
|  Pix2Pix-zero     |     25 sec    |4 min*   |
|  PnP              |     10 sec    |40 sec**        |
|  Instruct Pix2Pix |     11 sec    |N/A            |

(*: generating 100 prompts with GPT, obtaining embedding vectors, **: storing feature vectors, queries, and key values)

The computation cost of our method remains comparable to other approaches, although the Jacobian approximation takes around 2.5 seconds for $n=1$. This is because we only need to identify the latent basis vector once at a specific timestep. Furthermore, our approach does not require additional preprocessing steps like generating 100 prompts with GPT and obtaining embedding vectors (as in Pix2Pix-zero), or storing feature vectors, queries, and key values (as in PnP). Our method also does not require finetuning (as in Instruct Pix2Pix). This leads to a significantly reduced total editing process time in comparison to other methods.

**References**

[SDEdit] : SDEdit: Guided Image Synthesis and Editing with Stochastic Differential Equations, Meng et al., 2021

[Pix2Pix-zero] : Zero-shot Image-to-Image Translation, Parmar et al., 2023

[PnP] : Plug-and-Play Diffusion Features for Text-Driven Image-to-Image Translation, Tumanyan et al., 2022

[Instruct Pix2Pix] : InstructPix2Pix: Learning to Follow Image Editing Instructions, Brooks et al., 2022

---

### Author Response · Authors · 2023-08-16

We sincerely appreciate the diligent reviewers and AC for their efforts. We would like to kindly ask for missing responses for our initial rebuttal: cKEc, vuQy, and iEgM.

In addition, we notice that some reviews are considering our paper as just another diffusion-based editing method neglecting important aspects of our method. We would like to highlight our mathematical and contextual importance:
1. We provide *geometric interpretation of the latent space* and feature space (while the intricate interplay between semantics and the geometric structure within the latent space of diffusion models remains unexplored).
2. We introduce the first method that enables *unsupervised* image editing in both conditional and unconditional models within diffusion models (while contemporary works consider only unconditional models).
3. We show the first instance where semantic editing is possible with just *a single edit at a specific timestep* (while all other editing methods in diffusion model need multiple timesteps).
4. We propose the first method of *directly editing the latent space $x_t$* of the diffusion models.
5. Last but not least, we experimentally demonstrate the *characteristics of the diffusion model by analyzing its basis*.

We believe that our paper will significantly enrich future research.

---

### Decision · Program_Chairs · 2023-09-21

**Decision:**

Accept (poster)

**Comment:**

This paper utilizes a Riemannian pullback metric to analyze the latent space of diffusion models, utilizing this structure to demonstrate how the latent structure changes over time in the generation process, as well as how the latent structure changes under text conditioning in Stable Diffusion. This gives a form of interpretability of the diffusion model as well as a new understanding of the effect of its standard operations. The employed mathematical method also gives new insights.

The reviewers do have several concerns, most notably clarity and computational cost. The latter is perhaps less crucial for a post hoc interpretation method such as this one, but the authors would indeed need to use the feedback from the reviews to improve clarity. The authors should also carefully discuss the limitations of their method, including those brought forward in the review process.

In the post-rebuttal period, one reviewer steps forward as a champion of the paper, in particular arguing

"I like to see this kind of exploratory papers, since they could lead to a better understanding of the diffusion models and possibly other interesting applications. In particular, in this paper, the observations made about the geometric properties of the diffusion latent space are exploited in a novel (unsupervised) semantic editing, which is probably not as good as SOTA-supervised techniques, but it is still useful to validate the observed properties and appreciate the possible applications."

The AC agrees with this reviewer's opinion, and therefore recommends acceptance. The authors should note that this was a borderline paper, and they should carefully utilize the review feedback to improve clarity and downstream impact.